# IoT-Based Smart Irrigation Systems: An Overview on the Recent Trends on Sensors and IoT Systems for Irrigation in Precision Agriculture

**DOI:** 10.3390/s20041042

**Published:** 2020-02-14

**Authors:** Laura García, Lorena Parra, Jose M. Jimenez, Jaime Lloret, Pascal Lorenz

**Affiliations:** 1Instituto de Investigación para la Gestión Integrada de zonas Costeras, Universitat Politècnica de València, 46730 Grau de Gandia, Spain; laugarg2@teleco.upv.es (L.G.); loparbo@doctor.upv.es (L.P.); jojiher@dcom.upv.es (J.M.J.); 2Network and Telecommunication Research Group, University of Haute Alsace, 34 rue du Grillenbreit, 68008 Colmar, France; lorenz@ieee.org

**Keywords:** irrigation, IoT, precision agriculture, sensors

## Abstract

Water management is paramount in countries with water scarcity. This also affects agriculture, as a large amount of water is dedicated to that use. The possible consequences of global warming lead to the consideration of creating water adaptation measures to ensure the availability of water for food production and consumption. Thus, studies aimed at saving water usage in the irrigation process have increased over the years. Typical commercial sensors for agriculture irrigation systems are very expensive, making it impossible for smaller farmers to implement this type of system. However, manufacturers are currently offering low-cost sensors that can be connected to nodes to implement affordable systems for irrigation management and agriculture monitoring. Due to the recent advances in IoT and WSN technologies that can be applied in the development of these systems, we present a survey aimed at summarizing the current state of the art regarding smart irrigation systems. We determine the parameters that are monitored in irrigation systems regarding water quantity and quality, soil characteristics and weather conditions. We provide an overview of the most utilized nodes and wireless technologies. Lastly, we will discuss the challenges and the best practices for the implementation of sensor-based irrigation systems.

## 1. Introduction

The lack of fresh water is a rising concern, particularly in the Mediterranean countries or southern Asian countries such as India. Among the countries in Europe, the Mediterranean countries are the most vulnerable to drought [1]. A connection has been established between climate policies and water management. Water management can be affected by different variables such as the water demand from the different sectors or the consequences of some degrees of warming on hydrological resources. Climate change and its effects are a recurrent topic in research papers regarding water resources and agriculture. The possible consequences of global warming have led to the consideration of creating water adaptation measures to ensure the availability of water for food production and people and to maintain ecosystems [2]. Furthermore, the safety of the water to be consumed by humans and to be returned to the environment must be ensured. The possible risks of climate change are an increase in water shortage, the reduction of water quality, the increase in water and soil salinity, the biodiversity loss, the increase in irrigation requirements or the possible cost of emergency and remediation actions. These reasons have led to an increase in the number of studies focused on reducing water usage in irrigation processes. Some of these studies suggest the implementation of social, economic and climate change policies, as well as the implementation of technological innovations to improve water management.

The agricultural sector is one of the most important economic resources in these countries adding to the importance of managing well the available water resources to ensure the continuing of this economical sector. In India, 10% of the area of the country is covered by rice plantations [3]. Furthermore, 20% of the Indian population is below poverty levels and 15% is food insecure. Therefore, low food production affects both the population and the economy. In 2002, the monsoon season produced the least amount of precipitation in the last 130 years. That resulted in a loss of rice production due to the lack of fresh water. To determine the drought caused by anomalies in surface water, the Standardized Precipitation Evapotranspiration Index (SPEI) was utilized. These indexes and the information gathered from sensors that monitor the environment, the soil and the water can be utilized to determine the current state of the water and the possibility of covering all the freshwater needs. Countries with higher funds are already implementing systems for water management and water reuse aiming to optimize water usage and reduce the environmental impact caused by utilizing great amounts of water. However, some countries may find these solutions to be costly. 

Commercial sensors for systems aimed for agriculture and its irrigation are very expensive, making it impossible for smaller farmers to implement this type of system on their farms. However, manufacturers are currently offering low-cost sensors that can be connected to nodes to implement low-cost systems for irrigation management and agriculture monitoring. Furthermore, due to the interest in low-cost sensors for monitoring agriculture and water, new low-cost sensors are being proposed in researches such as a leaf water stress monitoring sensor [4], a multi-level soil moisture sensor comprised of copper rings placed along a PVC pipe [5], a water salinity monitoring sensor made with copper coils [6] or a water turbidity sensor made with colored and infrared led emitters and receptors [7].

Due to the recent advances in sensors for the implementation of irrigation systems for agriculture and the evolution of WSN and IoT technologies that can be applied in the development of these systems, we present a survey aimed at summarizing the current state of the art regarding smart irrigation systems. In this survey, we are going to provide an overview of the state of the research regarding irrigation systems. We will determine the parameters that are monitored in irrigation systems regarding water quantity and quality, soil characteristics, weather conditions, and fertilizer usage. We will provide an overview of the most utilized nodes and wireless technologies employed to implement WSN and IoT based smart irrigation systems. Lastly, we will discuss the challenges and the best practices for the implementation of sensor-based irrigation systems. 

Other authors have performed studies with a focus on irrigation systems, water management or precision agriculture systems. However, the other available surveys on smart irrigation systems analyzed quite a few papers [8,9,10,11,12] and therefore do not provide an in-depth analysis of the state of the art regarding irrigation systems. Others are focused on specific aspects regarding irrigation such as software for irrigation systems [13], pivot-center specific irrigation systems [14] or irrigation systems for greenhouses [15]. Lastly, there are surveys that focus on precision agriculture [16,17,18,19,20,21], crop monitoring [22] and the agro-industrial and environmental fields of agriculture [23] that comment on irrigation agriculture. In this survey, we provide an overview of the current advances in irrigation systems and the utilized sensors and actuators. Furthermore, we provide discuss the most utilized nodes and the wireless technologies employed for the communication and transmission of the data gathered by the sensors. This way, with this work we address the current gap in literature with a survey that provides an overview of IoT-based smart irrigation systems.

The rest of the paper is organized as follows: Section 2 presents the methodology employed to perform the survey. Section 3 presents the water management techniques utilized in current studies, the parameters that are utilized to determine the irrigation schedule and the related actuators. The sensors and parameters considered for the soil monitoring aspects of the evaluated smart irrigation systems are depicted in Section 4. The most monitored weather parameters and the sensors that monitor them are presented in Section 5. Section 6 comments on the most utilized nodes for IoT and WSN irrigation systems and the most popular wireless communication technologies and cloud platforms. The discussion of the current trends regarding IoT crop irrigation systems is presented in Section 7. Lastly, the conclusion and future work are presented in Section 8.

## 2. Materials and Methods

In this section, the process followed to elaborate this paper is presented. To perform this study, the following research questions were considered: What are the current IoT solutions for smart irrigation for agriculture? What sensors, actuators, nodes, and wireless technologies are being utilized to develop IoT irrigation systems? Search engines and digital libraries were utilized by the authors to search manually for papers suitable for this survey. A total of 283 papers were obtained from Google Scholar [24], IEEE explore [25], Scopus [26] and the digital library of Sensors [27]. The keywords employed to obtain the total number of papers to be analyzed were IoT irrigation, IoT irrigation system, and smart irrigation. Furthermore, all papers were checked to ensure they included the keywords irrigation or water and IoT or smart in their content. 

In order to discern which papers to use, only papers written in English were considered. Furthermore, to observe the recent evolution in this field, the selected papers were published from the years 2014 to 2019 (both years inclusive). Finally, a total of 178 papers were utilized to compile this review. 

After the selection process, all papers were classified manually into IoT irrigation systems and architecture, protocol and sensor proposals that were not used to create the figures in this paper but discussed relevant information. The papers from the first group (160) were then analyzed to determine the sensors and actuators that were employed to develop the IoT irrigation system, the parameters that were monitored, the type of irrigation and agriculture, the type of node, the type of wireless technology and data visualization technique that was utilized to access the data and to manually choose the actions of the irrigation system. All the collected data has been then classified into different sections and converted into graphs and tables to provide a complete overview of the actual state of the art regarding IoT irrigation systems.

The distribution of the papers regarding the country of the first author is presented in Figure 1. As it can be seen, the countries that investigate IoT systems for irrigation are countries where agriculture is a major economic source. India is the country with the highest number of papers with a total of 92 papers, 57.5% of the total. China and Spain are tied with seven papers each. Costa Rica, Ecuador, Indonesia Thailand, and the USA have between three and six papers. The rest of the countries that have investigated IoT systems for irrigation have one or two papers. 

It is remarkable that only one out of the five countries with the highest agricultural land area (China, USA, Australia, Brazil, and Kazakhstan) is included in the top five countries of our survey in terms of water management. India, which is the country with the highest production of papers that presents IoT systems for water management is the 7th country in terms of agricultural land area. With regard to the top six countries, in terms of published papers with irrigation systems, three of them (India, China, and Spain) have regions with high and moderate water scarcity problems. Again, there are some countries with regions affected by water scarcity that use irrigation systems in their agriculture and have published fewer papers on irrigation systems like the USA. Nonetheless, it does not mean that in these countries no efforts are done in order to reduce water use in agriculture. It is possible that in these countries the major efforts are done by enterprises and their findings are directly patented and distributed in the marked. Most of the papers are published by authors who come from developing countries. 

As per the number of papers per year of publication, see Figure 2, the interest in this topic has been increasing over the years. The lower amount of papers for 2019 is due to the year not being finished when the selection process of the papers was completed. Thus, not all the papers performed in 2019 had been published.

## 3. Water Management

In this section, we present the information and analysis of papers that present different techniques for water management in the irrigation process. For this analysis, we consider the papers that include any sort of water pumping actuator as an irrigation system. From a total of 178 evaluated papers, 107 of them present an actuator for irrigation. After analyzing the 107 papers we discard the ones that just offer partial information, a total of 89 papers are included in this section. 

In the agriculture activities that use water inputs, also known as irrigated agriculture, there are different manners to distribute the water. The different options present different efficiency and, in some cases, a specific manner should be used for a specific crop. The specific manners to irrigate have a great variety but we can divide them into the following categories: Attending to the way of water is distributed we can consider: (i) flood irrigation, (ii) spray irrigation, (iii) drip irrigation, and (iv) nebulizer irrigation. Regarding the existence of sensing systems we can have: (i) irrigation without any consideration, when the amount of water is not calculated or estimated, (ii) scheduled irrigation, when the water is supplied according to the estimated needs in a period of year, (iii) Ad hoc irrigation, when the amount of water is calculated based on the sensors measurements. The vast majority of the papers included in this section propose to use pumps and valves to distribute the water in conjunction with sensors to measure environmental parameters in order to calculate the water needs. From the 89 evaluated papers in this section, 83 include clear information of the proposed irrigation system, the other six only mention that they include actuators for irrigation, see Figure 3. Those 83 include different levels of detail, there are 49 papers that only indicate that there are motor/pumps in their system (40 paper) or valves (nine papers) without more detail. From those papers which offer more details, 19 of them include sprinklers (the most used system) [28,29,30,31,32,33,34,35,36,37,38,39,40,41,42,43,44,45,46], eight use drip irrigation [4,47,48,49,50,51,52,53,54], two propose the utilization of sprayers [4,55], and the rest use a very specific irrigation systems (robots [56], pivot [14], rain gun [57] or it can be applied to multiple systems [58]). In conjunction with the principal irrigation system, three papers propose the use of a fogging system [41,43,51] and two papers propose the use of fertigation in their systems [42,52]. 

In this paragraph, we describe the different agriculture systems that are included in the papers. Most of the included papers in this section do not describe the target agriculture system of their proposal. Nevertheless, there are 38 papers that include this information, see Figure 4. The most common use of irrigation systems is in outdoor agriculture, 50% of the proposed systems are for outdoor agriculture. Among this outdoor agriculture cases [14,29,31,32,38,40,42,46,47,48,52,59,60,61,62,63,64,65,66,67], some papers specify the agriculture products in more detail: general cereal [14], rice [48,59], spinach, beans, carrots, walnuts, corn, barley and maize [47], and multi-height fields [60]. In addition, among the 14 papers about greenhouses [28,33,39,41,45,51,68,69,70,71,72,73,74,75], five of them specify the type of crop. There are three cases of hydroponic crops [33,68,69], one case of mushroom cultivation [41] and one paper on flower farming [71]. Finally, there are five papers focused on gardening [36,53,55,67,76], one of them specifically about green walls [76]. 

Following, we describe the sensors used in the irrigation systems. From a total of 89 papers that include irrigation systems, there are six of them that do not describe or give information about the utilized sensors for their system. Those six papers are more focused on the node, telecommunication or visualization aspects. As in this survey we include several parameters, we will divide this information into two figures.

In the first graph, we are going to show what environments (air, water, soil or plant) have been monitored in most papers. This information is presented in Figure 5. A paper that monitored the soil can measure one or more parameters, but this information is not included in Figure 5. The environment that has been monitored in more papers is the soil. It was measured in 76 papers, more than 85% of the cases. Weather is the second parameter in terms of relevance of monitoring, it includes many parameters such as temperature, rain, and humidity among others. Weather is monitored in 60 papers, up to 68% of the proposals. Water is less measured in the proposals of irrigation systems. It is measured only in 26 papers, less than 30% of the included cases. The plant monitoring is the less measured factor. It is monitored only in four proposals. There are 15 papers that use the weather forecast to obtain data and include this data in their system. 

Following, the details about the number of monitored parameters are presented (see Figure 6). From the 89 included papers, more than half of the proposals measure between three and four parameters. Measuring fewer parameters (one or two) represents 31% of the cases (17% one parameter and 14% two parameters). There are some rare cases (10% of them) where more than four parameters are monitored. The papers where more parameters are measured are [64,67], with six measured parameters and [31,45,46,61,68,75,76] that measured five parameters. On the other hand, there are five papers (6% of cases) that do not measure any parameter [33,41,77,78,79]. 

Next, the details about the measured parameters are shown. Regarding the sensors of soil, plants, and water (see Figure 7), the most used sensor in the systems is the soil moisture sensor, it is used in 76 papers (more details in Section 4). First, we will analyze the data from soil and plant sensors. The soil temperature is much less considered in these systems, it is only measured in seven proposals. Attending to the soil characteristics, in one proposal the authors use a sensor that monitored the nutrients of the soil [80]. There are 14 proposals of IoT for irrigation management that include a pH sensor. However, not all of them specify if the pH sensors are for the soil or for the water [31,42,49,54,68,70,74,75,81,82], only four of them indicate this (three for soil [66,67,72] and one for water [69]). Only four papers consider plant monitoring, in two papers the measured parameter is the plant height [64,83], using an ultrasound sensor. In the other two cases, the leaf wetness was considered in their system [46,84,85]. Different sensors are used, [84] used a sensor based on optical signals while [46] utilized a commercial FC-37 sensor (Lydia Vogler, Peißenberg, Germany). 

Now, the data from the water sensor are presented. In this case, we will consider for the commentaries not only the data of the graphic (proposals with irrigation system) but also the information of the paper which measures the water quality, having or not the actuators for the irrigation system. The water level in the tanks is measured in 17 [32,36,51,57,58,61,62,64,69,85,86,87,88,89,90,91,92] out of 88 papers that have a pumping system. There are different methods to monitor the water level in the tanks and the most used is the one based on ultrasound [32,69,89], the resistive methodology is used in one case [36], the rest of cases do not offer information of how the data is measured. In the case of sensors based on ultrasound methods, the most utilized sensor is the HC-SR04 (Shenzhenshi guoyun dianzishangwu youxiangongsi, Shenzhen, China, used in [69]). In addition, there are six papers that do not include an actuator for the irrigation system even when they have water level sensors [60,93,94,95,96,97]. There are nine papers where sensors for monitoring water flow are utilized [14,31,42,45,50,64,75,76,82]. The sensors used are only described in two cases, the commercial Gems FT110 G3/8 sensor (Gems Sensors and Controls, Mumbai, India) is used in [75] and the YF-S402 (Wuhan yingying yingying xinxi jishu youxian gongsi, Wuhan, China) in the proposal presented in [76]. Both sensors are based on a Hall effect turbine. They have different prices and similar operational ranges. The Gems FT110 G3/8 can measure flows between 0.5 to 5 L/min. The other sensor, the YF-S402, is capable of measuring flows between 0.3 to 6 L/min. Moreover, water flow sensors are used in three other proposals that do not include actuators for irrigation systems [5,98,99]. Regarding water quality, the water conductivity is monitored in four proposals [51,68,69,75]. Only in one case [75], the authors indicate the used sensor. They use commercial sensors for monitoring pH and conductivity from B&C Electronics (Carnate, Italy).

Additionally, there are two papers without an actuator for irrigation systems that include in their proposals a sensor for conductivity [98,100]. Nevertheless, no data regarding the equipment used are given. Regarding the pH of water, it is monitored in another three papers (apart from [69]) [99,101,102]. In one case [99], the authors specify the used sensor, the Lutron Pe-03 (Lutron, Taipei, Taiwan). The temperature of the water is measured in two proposals [51,69]. Only in one case [69] did the authors specify that the used sensor is the DS18B20 (Adafruit Industries, New York, NY, USA). No other proposals evaluated in this survey (having or not irrigation system) presented the use of temperature sensors for the water. 

On the other hand, with regard to the proposals that monitored the atmospheric parameter, the data is presented in Figure 8. Next, we are presenting the importance of these sensors for irrigation systems. No details of which sensors are used in which papers are given in this section. This information will be further discussed in Section 5. The air temperature is the most measured parameter, it is monitored in 58 papers. A total of 51 proposals include a humidity sensor. Other parameters are less measured such as the rain, which is measured in eight out of 89 papers that proposed an irrigation system, or the wind, measured in only one paper. Meanwhile, there are 15 proposals that consider the weather forecast. From those 15 proposals, 10 of them use the data from the forecast in conjunction with the data gathered by sensors and five proposals use the weather forecast as the sole information for weather data/atmospheric parameters.

Next, the information related to the use of actuators in the papers that proposes an irrigation system included in this section is listed in this paragraph (see Figure 9). 

The vast majority of the papers include only the actuator of the irrigation system (pump, valve, sprinklers, etc.). A total of 71 proposals (81% of cases) use only the irrigation system as an actuator. There are seven papers (8% of cases) that propose the use of two actuators (irrigation system plus another one). The use of three actuators is presented in seven papers, the same percentage that the use of two actuators. Finally, there are some rare cases (3%) where authors have proposed the use of more than three actuators, in two papers they use four actuators [39,72], and in one case the use of five actuators is presented [67]. 

Next, we present in Figure 10 the description of the selected actuators. Since we are considering only the papers that include an irrigation system, the most used actuator in the different irrigation systems. Therefore, we do not include the use of irrigation actuators in Figure 10. The most utilized actuator is the sound emitter, used in seven cases [32,49,58,59,61,66,103]. It is generally used as an alarm for the farmers or as a dissuasive measure to prevent the entry of animals into the farming lands. In second position, we found the use of artificial light systems. They are used in six cases [36,38,50,71,72,75]. The use of artificial light is mainly related to the generation of artificial photo- periods to increase productivity, but they can also be used for operational purposes. Fans are included in five proposals and their use is related to modifying the atmospheric conditions (temperature, humidity or even the balance between CO_2_/O_2_). The proposals that include the use of fans are [39,65,68,72,75]. The use of ultrasonic sound emitter is found in four papers [32,51,64,65]. The least utilized actuators in IoT irrigation systems are scarecrows [58,65,66], which are dummies or robots that are utilized or have the function of scaring birds and other types of animals to protect the crops, and heaters [28,65], that modify the temperature to improve the performance or the health of the crops.

In addition to the defined parameters, which are measured in different papers included in this survey, there is an important parameter that is crucial for irrigation scheduling. This parameter indicates the water loss due to the evaporation from the soil and transpiration through the stomata of plants; it is known as evapotranspiration (ET). Several authors measured both terms—evaporation and transpiration—separately, using different sensors to obtain the ET data [104]. It also can be estimated according to the vegetation and the climatic data by several mathematic models [105]. Some web-based applications can estimate the potential ET given a location. For ET monitoring, remote sensing is the best option as many papers point out [106,107,108]. Among the possibilities for ET monitoring with sensors, the integration of soil moisture changes can offer evaporation data. Nonetheless, the measurement of transpiration is more complex; it can be estimated with the data of sensors that measure radiation, leaf area index, and vapor pressure deficit [109]. Due to the complexity of its measurement, it is not included as a monitored variable in most of the papers of precision agriculture. None of the papers considered by this survey includes all the necessary sensors for measuring the ET.

## 4. Soil Monitoring

An outline of the soil parameters considered for the IoT irrigation monitoring systems and the utilized sensors are detailed in this section. In this section, we include the data from those proposals that include at least one sensor for the soil, a total of 106 papers are included.

As indicated in the previous section, the most relevant parameter for the irrigation systems is soil moisture. The number of proposals that include different sensors is presented in Figure 11. The soil moisture is measured in all the 106 papers included in this section. The second parameter, in terms of relevance, is the soil temperature. The soil temperature is monitored in nine proposals. The pH is monitored in four papers [52,66,67,110] and the nutrients of soil in only three cases [5,80,102].

From the 106 proposals that include a soil moisture probe, 71 of them do not indicate the used equipment to measure the soil moisture, see Figure 12. In six cases [38,101,111,112,113,114], they only include a picture of the device, but they do not indicate the model and manufacturer. In all those cases, the sensors employed are based on the measurement of the conductivity between two electrodes which are inserted in the soil. In one case [94], the authors proposed to use a 4-fork sensor, based on the same principle as the previous ones. Nonetheless, no extra information about the sensor used was included in the paper, such as the manufacturer or model. The proposal presented in [98] only indicates that the utilized sensor was manufactured by EB. In [64] the authors only mention that the sensor used was a hygrometer. A total of 24 proposals clearly indicate the used sensor for monitoring the soil moisture. 

Most of those proposals use similar sensors based on the conductivity between two electrodes. Different models of the aforementioned sensor have been used in the proposals (see Figure 13). The most used sensor, based on conductivity, is the YL69 (SparkFun Electronics, Niwot, CO, USA). It has been used in nine proposals [44,56,71,87,95,115,116,117,118]. This sensor is characterized by a low price and it is created specifically to operate with Arduino (and similar nodes). The output voltage goes from 3.3 to 5 V and the output values are related to soil moisture (0 to 300 for dry soil, 300 to 700 humid soil). Nevertheless, it is not possible to have a measure that indicates exactly the value of moisture without performing new tests and calibration of the sensor. The same range of measure is found in the other used sensor based on the same principle, the FC-28 (Uruktech, Baghdad, Iraq, used in [34,72,76]) and the SEN0114 (DFRobot, Shanghai, China, used in [46]). The S-XNQ-04 sensor (XNQ Electric Company Store, Beijing, China), used in [55] has a similar operation principle. However, it is composed of three electrodes. In this case, the sensor output voltage goes from 0 to 2 V and the sensor is capable of measuring the soil moisture giving as a response the relative saturation moisture content expressed in percentage (%). This sensor has an accuracy of 3% and it can measure from 0 to 100% of relative saturation. It can operate from −40 to 85 °C. 

A device able to measure the temperature and the moisture at the same time, the VH400 (Vegetronix, Inc., Riverton, UT, USA), is utilized in [37,40,89,96,119,120]. These sensors can be used from 40 °C to 85 °C, but the accuracy and operational range of the sensor probe are not indicated in its datasheet. The operational principle of this sensor is based on the measurement of the dielectric constant of the soil using transmission line techniques to measure the moisture in any type of soil notwithstanding soil salinity. The output voltage of this sensor goes from 0 to 3 V. This output voltage is related to the volumetric water content in the soil. The sensor does not offer a linear relationship between the output voltage and the volumetric water content, but the sensor can measure from 0 to up to 60% of volumetric water content in the soil. This sensor has a relatively low price and can be implemented in Arduino systems (and similar). 

A sensor with completely different operational principles, the 200SS (HYPERION TECHNOLOGIES B.V., Delft, The Netherlands), is used in two proposals [14,121]. It is a solid-state electrical resistance sensing device that is used to measure soil water tension. The 200SS has a pair of highly corrosion-resistant electrodes, as a resistive device, the resistance of the sensor changes with the soil moisture. The electrodes are embedded within a granular matrix. The output values of the sensor go from 0 to 199 centibars. The higher the value of centibars the lower the water availability for the plants. The system presented in [54] uses an SM300 sensor. This device is able to measure two soil parameters (moisture and temperature). The SM300 measures the soil moisture and temperature based on the generation and propagation of the electromagnetic field into the soil. The sensors measure the soil moisture as the volumetric water content, the operating range goes from 0 to 100% and the accuracy is ±2.5%. This sensor is manufactured by Delta-T Devices Ltd. (Cambridge, UK). This sensor is not specifically created to be integrated in an Arduino measuring system, even though it can be adapted since the output voltage goes from 0 to 1 V. 

Finally, the system detailed in [122] used the HA2002 sensor, but no information is available on the internet about this sensor in terms of operational principle nor range of measuring. The sole information that was found was that the sensor is an HA2001 sensor manufactured by Handan Dingrui Electronics Co., Ltd. (Handan, China). 

Regarding the sensors used for monitoring the soil temperature, which are mentioned in [37,40,45,54,63,67,94,122,123], the different sensors are detailed in this paragraph. There are five papers that do not specify the sensors used [45,63,67,94,123]. The other four proposals indicate in the paper the sensors selected. Although in one paper no information could be found about the sensor utilized [122], the authors indicate that their system includes the HA2001 sensor.

In other cases, the information of the selected sensors is detailed. The proposal presented in [54] used an LM35 sensor. This device is able to measure temperature. The operating range is −40 to +125 °C and the accuracy is ±0.5 °C. The sensors offer an analogic signal and it is manufactured by Texas Instruments (Dallas, TX, USA). The proposals presented in [37,40] selected the THERM200 (Vegetronix, Inc., Riverton, UT, USA) to measure the temperature. The operational range of these sensors goes from −40 °C to 85 °C and its accuracy is ±0.5 °C. The operation principle of this sensor is not specified in the datasheet. 

In terms of sensors for soil nutrients, three papers include in their proposals sensors for monitoring this variable [5,80,102]; nonetheless, not all of them indicate the used sensor. The only paper that clearly indicates the utilized sensor is the proposal presented in [80] includes an 1185 SunRom color sensor (Sunrom Electronics, Ahmedabad, India). This sensor is based on RGB color detection, in [80] authors relate the color of the soil with the nutrients. The pH of the soil is measured in four proposals [66,67,72,110]. However, none of these proposals indicates the selected device or equipment in the paper. 

## 5. Weather Monitoring

An overview of the weather parameters monitored in IoT irrigation monitoring systems and the most utilized sensors to monitor these parameters is provided in this section. The weather conditions are a key factor both in irrigation needs and in the performance of crops. Figure 14 shows the most monitored weather parameters in the currently available smart irrigation proposals. Temperature and humidity affect the evapotranspiration of the water in the soil. Air temperature is defined as the thermic level of the atmosphere and is usually measured in Celsius, Fahrenheit or Kelvin degrees. It is the most monitored weather parameter with one hundred papers monitoring it. Humidity is the presence of water vapor in the air. The amount of water vapor in the air is expressed in percentage (%). It is the second most monitored weather parameter with 86 papers. Furthermore, luminosity is paired with temperature as direct radiation from the Sun raises the temperature and leads to more water loss in the soil. It is defined as the intensity or brightness of the light. It is measured in Lux. It is the third most monitored parameter with 28 papers. The amount of precipitation determines whether the irrigation is needed or not and the amount of water to be employed in the case of irrigation being necessary. It considers any form of hydrometeor that falls from the atmosphere and reaches the earth’s surface. Furthermore, it includes rain and snow. A total of 14 papers considered precipitation monitoring. Although temperature, relative humidity, luminosity, and precipitations are the most important factors regarding IoT irrigation systems for agriculture, many systems propose the monitoring of other environmental parameters. Table 1 details all the papers that utilized these parameters.

As it can be seen, the four previously mentioned weather parameters are the most recurrent environmental factors in the literature. However, there are other weather factors concerning the air and climate that are monitored in some IoT smart irrigation systems. Ultraviolet radiation is electromagnetic radiation with wavelengths ranging from 10 nm to 400 nm and is monitored by seven papers [14,66,68,75,120,138,145]. Measuring wind speed [14,45,46,67,72], wind direction [45,46,67] and atmospheric pressure [158] is also considered by some papers. Air pollutants are considered in some IoT irrigation systems as well. CO_2_ [45,63,66,67,75,138,145], ozone [102], gas [95,143] and smoke [124] are the types of air pollutant considered by the currently available smart irrigation systems. Lastly, other parameters that are monitored concerning the surrounding area are movement and sound. Movement [38,42,58,61,65,127,141,143,146,148,163] and sound [42,138,140,163] are considered mostly for security purposes. Ultrasound sensors are deployed to detect intruders in the fields. 

There are varied sensors that can be utilized to measure weather parameters. However, there are smart irrigation systems that opt for not including these sensors and choose to ask for the environmental information to web servers such as Yahoo [62,127], that provide the weather and environmental data from meteorology agencies.

Temperature monitoring is one of the most common parameters regarding weather monitoring in IoT irrigation systems. As it can be seen in Figure 15, The DHT11 (Adafruit Industries, New York, NY, USA), DHT22 (Adafruit Industries, New York, NY, USA) and the LM35 (Texas Instruments, Dallas, TX, USA) are the most utilized temperature sensors with 27, 13 and 14 papers respectively. Al these sensors are low-cost sensors. On the one hand, the DHT11 and DHT22 provide both temperature and relative humidity readings. On the other hand, the LM35 and the TMP-36 (Analog Devices, Norwood, MA, USA) have broader temperature ranges. The papers that utilized these sensors are detailed in Table 2. These sensors are low-cost which may be relevant to their extended use. 

According to sensor type, thermistors such as the DHT11, the DHT22 and the AM2315 (Adafruit Industries, New York, NY, USA) are the most utilized type of temperature sensors, [14] (see Figure 16). The DHT11 and DHT22 are low-cost sensors. However, the AM2315 has a higher price with the advantages of a higher range (−40 °C to 125 °C) has higher accuracy (±0.1 °C). Semiconductor-based integrated circuit temperature sensors like the LM35 (Texas Instruments, Dallas, TX, USA) and DS18B20 (Maxim Integrated, San Jose, CA, USA) [50], and band-gap proportional to absolute temperature sensors, SH10 (Sensirion AG, Staefa ZH, Switzerland) [35] and SH11 (Sensirion AG, Staefa ZH, Switzerland) [46], are the next most utilized temperature sensors by type. The SH10 and SH11 sensors provide both temperature and relative humidity readings. Furthermore, compared to the DHT11, they present better temperature ranges and accuracy. They are however similar to the DHT22 in terms of temperature ranges, accuracy and price. The TMP36 (Analog Devices, Norwood, MA, USA) [34,152] is a temperature sensor comprised of a diode and the EE160 (SENSOVANT, Paterna, Spain) [75] is a resistive temperature detector sensor. The FM-KWS (Dongguan Holchan Electronics Technology Co., Ltd., Dongguan, China) [122] and the BME250 (Adafruit Industries, New York, NY, USA) [157] are of an unknown type.

The most utilized relative humidity monitoring sensors are presented in Figure 17. As in temperature monitoring, the DHT11 and DHT22 sensors are the most utilized ones, with 29 and 13 papers that use them, respectively. As both sensors are able to monitor both temperature and humidity, most systems that utilize these sensors do not choose a separate sensor for each parameter. There are sensors with better accuracy. However, IoT irrigation systems generally do not need the highest accuracy levels available in the market. Table 3 presents the papers that utilize these sensors.

According to type (see Figure 18), the most utilized humidity sensors for IoT irrigation systems is the capacitive humidity sensor, which are the DHT11, the DHT22, the AM2315 [14], the SH10 (Sensirion AG, Staefa ZH, Switzerland) [35], the HIH 4000 series (Honeywell International Inc., Charlotte, NC, USA) [98] and the HH10D [166]. The DHT22, AM2315, and SHT10 have the same range. However, the AM2315 has higher accuracy (±2%) than that of the DHT22 and the SH10 (±5% and ±4.5% respectively). Resistive humidity sensors are another one of the types of available humidity sensors, such as the HR202 [48]. However, there are five humidity sensors employed for IoT irrigation systems that are of an unknown type. They are the FM-KWS (Dongguan Holchan Electronics Technology Co., Ltd., Dongguan, China) [122], the BME280 (Bosch, Gerlingen, Germany) [157], the EE160 (SENSOVANT, Paterna, Spain) [75], the HTU210 (Adafruit Industries, New York, NY, USA) [46], and the SY-HS-220 (SYHITECH, South Korea) [49].

Light intensity and UV radiation are another set of parameters that are frequently monitored in IoT irrigation systems. Figure 19 presents the most utilized sensors for light intensity monitoring. The light dependent resistor (LDR) is the most utilized sensor for light intensity monitoring [28,48,72,76,84,119,129,131]. The second most used luminosity sensor is the BH1750 (ROHM Apollo Co., Ltd., Fukuoka, Japan) [62,126,146,157] followed by the TSL2561 (TAOS Inc, Plano, Texas, USA) [145,158] with one paper less. The LS-BTA Vernier (Vernier, Beaverton, OR, USA) was used as well in [98]. Among them, the BHT1750 is the sensor with a broader range (1-65535 Lux). There is not a commonly utilized sensor for radiation, but the sensors currently utilized in IoT irrigation management systems are the SN-500 (Vernier, Beaverton, OR, USA) [14], the GUVA-S12SD (ROITHNER LaserTechnik, Vienna, Austria) [120], the 6450 TSR (Davis Instruments Corporation, Hayward, CA, USA) [145] and the SP110 [75].

Regarding rain, one of the ways in which it is incorporated in the decision-making process of irrigation systems is to stop automatic irrigation if rain is detected to avoid overwatering [42]. The RSM3ALS sensor was utilized in [119]. It is an optical rain sensor comprised of an LED, an LRD, an electronic control unit, lenses and an ambient light sensor. When the sensitive area is dry, a high intensity of light is received by the IR receiver. When the sensitive area is wet, the intensity of the received IR light decreases. It is manufactured by Bosch (BOSCH, Gerlingen, Germany) and it is commonly used in the automotive industry. The SEN-08942 is a meteorology station manufactured by Sparkfun (SparkFun Electronics, Niwot, CO, USA). It allows monitoring both wind speed and direction and it incorporates a pluviometer [46]. The PRD180 is comprised of a rain detecting board with two separate PCB tracks and a control module that provides a digital and an analog output [48]. The operating voltage is from 3.3 V to 5 V. It is manufactured by Elecmake (India). Like the PRD180, the YL83 (Vaisala, Vantaa, Finland) is comprised of a series of conductive strands printed onto a Bakelite plaque [76]. The water creates a short-circuit offering a low resistance between the lines connected to the ground and the lines connected to the positive polarity.

The type of agriculture most concerned with air and wind as a parameter concerning weather is greenhouse agriculture. For example, one of the aims of [68] is the sanitation of the air of the greenhouse from pathogens and chemical contaminants. This is then closely related to the activation of the actuators necessary to the regulation of the quality of the air. The contaminants that can affect the quality of the air are CO, CO_2_, O_3_, NO_x_, PM_2.5_, and PM_10_, among others [64]. In the context of agriculture, air contaminants are monitored with the use of sensors that can be low-cost sensors such as the MQ135 (SHENZHEN INVENTOR ELECTRONIC TECHNOLOGYCO., LTD., Shenzhen, China), MQ131, MQ2 or MQ9 sensors or commercial sensors that can be found in professional weather stations. The MQ135 [64,143], used for benzene, alcohol, smoke, NH_3_, NO_x_ and CO_2_ monitoring, and the MQ2 [64,95], used for gas monitoring (propane, hydrogen, and LPG), are the most utilized low-cost sensors for air quality monitoring. The MQ131 is utilized for ozone, Cl_2_, NO_2_ monitoring [64] and the MQ9 is utilized for CO/combustible gas, methane, carbon monoxide and LPG monitoring [64]. Another sensor that can be used to monitor CO_2_ is the CDM4161A (Toshniwal Sensing Device Private Limited, Ajmer, India) [145]. 

Wind speed and direction may be of interest to IoT irrigation systems such as for the proposal in [46], that utilized the SEN-08942 sensor manufactured by Sparkfun. Barometric pressure can be of interest as well and can be measured by the BME280 sensor manufactured by Bosch [157].

Movement detection is often related to security measures or to the aim of scaring animals to avoid crop damage. The most utilized sensor for movement detection is a passive infrared (PIR) sensor [42,95]. This sensor measures the infrared light that radiates around the objects placed at the front of the sensor. The change in infrared radiation is converted into changes in voltage that trigger the detection. It is often utilized for intrusion and pest detection. Ultrasonic sensors are often utilized to detect movement, such as the robot proposed in [65] that is able to perform some actions to scare the animals that may enter the fields. These sensors are comprised of a transmitter and a receiver that operate at the 40 kHz frequency. Its range is from 10 cm to 30 cm. Ultrasound at lower frequencies was utilized in [60] to repel the animals. Insects may cause great damage to the crops, for that reason some systems such as [45] include a pest warning functionality. The system in [133] is able to even detect the type of insect pests that are damaging the crops. These features are often accompanied by a buzzer to generate an alarm when intruders or animals are detected [61]. 

Depending on the type of agriculture, the values from the measures for each environmental parameter can be utilized to determine the actions of different types of actuators. It can be part of a model like the one proposed in [72] where the on-agriculture stage gathers the data on temperature, humidity, light, moisture and wind speed in order to activate or deactivate fans, artificial light, a cooler and water pumps. Many IoT irrigation systems have both a manual and an automatic mode to control irrigation. In [64], the amount of water released by the water pumps or the duration of the irrigation can be selected in the manual mode. For the automatic mode, a preset value is utilized at the beginning. Then, the data is analyzed to determine the personalized settings. The most utilized actuators to modify the conditions of the climate that affects the crops or to protect the crops from animals and intruders are presented in Figure 20.

Buzzers are the most utilized actuators in IoT irrigation systems regarding external agents. These sound emitters are usually utilized to scare animals and to alert the user of anomalies in the system or the intrusion of animals or people [32,49,58,59,61,66,91,95,103]. Fans [39,65,68,72,75,152,166] and artificial lighting [39,51,71,72,75,82,154] are tied in first place with seven papers each. The fans can be utilized to cool the temperatures and to create a breeze or light wind to aid in the pollination process. Artificial light is utilized to aid plants in performing photosynthesis. Different colors may be good for different aspects of the plant growth such as blue for the leaves and a mix of red and blue for the flowers. Coolers are the third most employed actuator for weather conditions controlling, utilized for cooling the temperature and for aiding in the control of the relative humidity, with four papers [28,39,68,72]. The least utilized actuators in IoT irrigation systems are heaters [28,65], that modify the temperature to improve the performance or the health of the crops. Most of the actuators regarding weather conditions are usually placed inside greenhouses. Whereas buzzers and scarecrows can be placed in outdoor fields.

## 6. Sensor Networks for Irrigation Systems

In this section, an overview of the most utilized nodes for implementing sensor networks intended for irrigation systems is presented. Furthermore, the most employed wireless communication technologies are going to be presented as well. Lastly, the most frequent cloud systems for IoT in smart irrigation solutions and common architectures for these systems are going to be discussed as well.

### 6.1. IoT Nodes for Irrigation Systems

In this subsection, the most utilized IoT nodes for IoT irrigation systems are going to be presented. Figure 21 presents the most utilized nodes for the implementation of IoT irrigation systems. As it can be seen, Arduino boards (Smart Projects Srl, Scarmagno, Italy) are the most utilized nodes for the implementation of IoT irrigation systems. The Arduino UNO was utilized in 34 papers and the Arduino Mega was used in six papers. Furthermore, a total of 59 papers claimed to use an Arduino board. The references of the most utilized nodes are detailed in Table 4.

Figure 22 shows the types of Arduino nodes utilized in IoT irrigation systems. In addition to the Arduino UNO and the Arduino Mega, the Arduino Yun [60], the Arduino Due [96] and the Arduino NANO [151] were utilized in other IoT irrigation proposals. However, 15 papers did not specify the model of the Arduino board utilized [5,29,32,51,81,82,90,120,124,131,138,162,164,167,177].

Some other popular boards are manufactured by other companies but can be programmed utilizing the Arduino IDE. One of these boards is the Node MCU (Espressif Systems, Shanghai, P. R. China), which was utilized in 13 papers. The other boards are the Wemos MINI D1 (Wemos, P. R. China) and the Galileo Gen-2 (Intel, Santa Clara, California, USA) both utilized in three papers each.

The nodes from the Raspberry family (Sony, Pencoed, Wales) were frequently utilized as well. The most utilized Raspberry node was the Raspberry Pi 2 Model B and the Raspberry Pi 3 Model B+, with four papers and three papers, respectively (see Figure 23). They are followed by the Raspberry Pi 3 Model B [132,151], the Raspberry Pi 1 Model B [76,117] and the Raspberry Zero [152]. There were however eight papers that utilized an unspecified Raspberry Pi 3 [28,46,55,99,110,148,159,163] and 13 papers [39,50,59,65,96,115,120,124,134,135,136,172,178] that indicated that a Raspberry Pi node was utilized but the model of the node was not specified. The Raspberry Pi boards are more potent than the Arduino boards. Oftentimes, both Arduino and Raspberry Pi boards are utilized to implement an IoT irrigation system, using them according to the processing requirements of each task.

Furthermore, there are other less popular nodes utilized by other IoT irrigation proposals. These other nodes are the waspmote (Libelium Comunicaciones Distribuidas S.L., Zaragoza, Spain) [157], the crowduino (Elecrow, Shenzhen, China) [38,81], the LPC2387 (NXP Semiconductors, Eindhoven, The Netherlands) [135], the IPex16 from OdinS (Odin Solutions, S.L., Alcantarilla, Spain) [75], the Renesas (Renesas Technology, Tokyo, Japan) [78], the Mica (Harting Technologiegruppe, Espelkamp, Germany) mote [37], the eZ430-RF2500 (Texas Instruments, Dallas, Texas, USA) [179], the AESP ONE (Department of Innovation Engineering, University of Salento, Lecce, Italy) [94], the FORLINX OK6410 (Forlinx Embedded Tech. Co., Ltd., Baoding City, P. R China) [122], the BeagleBone Black (Waveshare Electronics, Shenzhen, P. R China) [95], the Edison (Intel, Santa Clara, California, USA) [29], the CC3200 Simple Link (Texas Instruments, Dallas, Texas, USA) [62,180] and the CC1310 (Texas Instruments, Dallas, Texas, USA) [147].

The selection of the best node for an IoT irrigation system will depend on the necessities and the characteristics the farmer wants for the system. Arduino nodes and similar nodes from other brands provide a low-cost solution that can be implemented in developing countries and smaller farms. On the other hand, Raspberries have powerful computing abilities that allow the implementation of more demanding software and algorithms.

Many works do not use a microcontroller board like the ones presented above and opt to implement their system with their circuit design. Figure 24 presents the most utilized processors for smart irrigation systems. The most utilized controllers are the ATmega328 [93,103,170,181,182] (Atmel, San José, California, USA) and the ATmega2560 [56,72,81] with five and three papers respectively. The Atmega1281 [133,157] and the LPC2148 (NXP Semiconductors, Eindhoven, The Netherlands) [48,123] are both utilized in two papers each. The other controllers comprise the LPC2138 [80], ATmega8 [101], the ATmega16/32 [65], the MSP430F5438A (Texas Instruments, Dallas, Texas, USA) [67], the MSP430F5419A [179], the MSP430F2274 [179], the STM32L151CB (STMicroelectronics, Geneva, Switzerland) [78], the STM32F205 [152] and the PIC16F877A (Microchip Technology Inc. (Microchip Technology), Chandler, AZ, USA) [42]. The proposals that opt for developing their own designs for the nodes aim at addressing their own particular requirements. Therefore, the selection of the processor would depend on the characteristics of the IoT irrigation system considering the type of crop and its irrigation needs.

### 6.2. Communication Technologies

In this subsection, an overview of the most employed wireless communication technologies is going to be presented as well.

According to the Machina research report, the number of connected agricultural devices is expected to grow from 13 million at the end of 2014 to 225 million by 2024 [183]. In addition, the report predicts that around half of those connections will use the newly emerging low power wide area network technologies which are particularly well suited to many of the applications in agriculture.

When IoT devices are implemented, the employed communication technologies are a key point to achieve successful operation. In a generic way, communication technologies can be classified according to the environment where they will be deployed, the utilized communication standard or the utilized spectrum band. Table 5 shows the main technologies used in IoT for irrigation systems.

If the scenario is considered, they can be classified into two types depending on the purpose of the IoT devices. On the one hand, we can distinguish the devices that function as nodes that transmit small amounts of data at short distances and have low energy consumption. On the other hand, there are devices that can transmit large amounts of data over long distances that have high energy consumption. Therefore, range, data rate, and energy consumption are some of the most important aspects to consider when deciding which technology to use.

If the utilized standard is considered, there are a large number of wireless standards that are used in the communications of IoT devices. A general classification can be established between devices that communicate at short or long distances. Short distance communication standards usually reach distances of up to 100 m. Long-distance communication standards reach distances of kilometers. 

Also, another type of classification can be established based on the employed communication technology. In this way, three categories can be established. The first category is formed by those that connect utilizing Ethernet or cable and WAN technology. The second category is comprised of those that connect via WiFi and WAN technology. Finally, the third category is formed by those that connect using both low power WiFi and WAN technology.

If the spectrum band used is considered, we can classify them into technologies that use the unlicensed spectrum or the licensed spectrum. Unlicensed spectrum bands are known as Industrial, Scientific, and Medical (ISM radio band bands, and are defined by article 1.15 of the International Telecommunication Union’s (ITU) ITU Radio Regulations (RR) [184]. The ISM bands can vary according to the region and its permits. Its main advantage is its ease of installation and the main drawbacks are related to safety deficiencies and possible interferences. The licensed spectrum bands are assigned according to the different proprietary technologies. They offer greater security, greater coverage, and fewer interferences. However, the cost of using a licensed band is higher. 

The most utilized communication technologies among the reviewed papers are presented in Table 6. In many of the proposals that we have studied, the authors combine several communication technologies instead of using one single protocol. As an example, it is very common for authors to start using either a wired or a wireless technology to connect the sensors with the nodes, and then, use a wireless technology to send data from nodes to remote or cloud storage centers. 

According to the descriptions provided by the different authors of their own implementations, we have verified that the most used communication technology in the different proposals is WiFi, as it can be seen in Figure 25. The reason could be due to is accessibility. The currently available low-cost devices for IoT usually support WiFi and, although its range can be considered short for the average expanse of a farm, small farms could be able to provide enough wireless coverage with several low-cost devices. GSM and ZigBee are widely spread wireless technologies as well, with 25 and 23 papers that use them respectively. GSM provides long-range communication at the cost of a mobile plan of the service provider that operates in the area. ZigBee provides low energy consumption and allows implementing networks with up to 65000 nodes. However, it has lower data rates than other available technologies and its range would imply the deployment of many nodes. Lastly, there are two new technologies that have been getting popular recently. LoRa is able to provide very long ranges, which makes this technology a very good option for secluded areas with no service. Moreover, regarding specific protocols for IoT systems, it is a little bit surprising that even though MQTT is a widely spread protocol due to its low power consumption and low overhead, it is not a popular protocol for IoT irrigation systems at the moment. It is observed that a relatively high number of papers, exactly in 24 papers, did not describe the employed technology. In Figure 25 this group has been named as Undefined.

We also observe that, in a large number of works, the end-users access the data, obtained during the monitoring process or to carry out the control of the system, through APPs or WEBs. Most of these communications are done through mobile devices using wireless technologies.

Table 7 presents the communication modules that have been used in the reviewed articles. It is observed that most of the authors have not indicated the communication module employed. Furthermore, it can be observed that the most utilized module is the ES8266 WiFi module (SparkFun Electronics, Boulder, CO, USA) with 29 papers. It is followed by the SIM900 GSM module (SIMCOM Wireless Solutions, Shanghai, China ) with nine papers and the NRF24L01 2.4 GHz module (Longruner, Shenzhen, China), the XBee S2 ZigBee module (Digi International Worldwide Headquarters, Hopkins, MN, USA) and the SX1276 LoRa module (Semtech, Camarillo, CA, USA) with three papers each. Some other less utilized modules and chips for wireless communication include GSM modules such as the SIM800 (SIMCOM Wireless Solutions, Shanghai, China) [171,187], the SIM300 (SIMCOM Wireless Solutions, Shanghai, China) [189], Ethernet modules such as the W5100 (SparkFun Electronics, Boulder, CO, USA) [72] or the Ethernet Shield (ELEGOO, Shenzhen, China ) utilized in [139], WiFi modules such as the ESP32 (Espressif Systems, Shanghai, China ) [119], the ESP1 [112], the ATWIN C1500 (Microchip Technology Inc., Chandler, AZ, USA) [191] and the Broadcom (Broadcom, San José, CA, USA) [152], LoRa modules such as the Feather 32u4 (Adafruit Industries, New York, NY, USA) [191] and the LoRa ESP32 [196], 2.4 GHz RF modules such as the C2500 (Mascot, Tainan City, Taiwan) [179] and sub-1 GHz RF modules such as the CC1310 (Texas Instruments, Dallas, TX, USA) [147] and the CC1101 [150], Bluetooth modules such as the HC-05 (Guangzhou HC Information Technology Co., Guangzhou, China) [129,198] and near radio frequency nodules such as the NRF4L (Shen zhen City Huo Chuang Technology Company Ltd., Shenzhen, China) [181]. There other modules that are compatible with several technologies such as the JN5139 that admits both WiFi and ZigBee [121] or the Dragino LoRa GPS shield (DRAGINO TECHNOLOGY CO., LIMITED, Shenzhen, China) [144].

### 6.3. Cloud Platforms

In this subsection, an overview of the most frequent cloud systems for IoT in smart irrigation solutions is going to be discussed. The two main storage systems used by the authors are traditional databases or cloud. In one article, the authors indicate that they store the data in a Raspberry Pi [124]. As it can be seen in Figure 26, a large percentage of the authors, more than 50%, have not defined the storage system they have used, there are 79 other papers where the storage system used is not specified. 

To be able to provide the new services that are currently demanded in IoT, it is necessary to use middleware. Through middleware, it is possible to connect programs that initially had not been developed to be connected to each other. Bandyopadhyay et al. [199] presented a classification of the IoT-middleware, based on the various features and interface protocol support. One of the reasons why middleware is needed to connect IoT devices, which are initially autonomous, is to provide cloud services. We can take advantage of existing middleware using it on different cloud platforms.

When we talk about the cloud, we identify a place where data, which is collected through sensors and transmitted to remote locations, will be stored and processed. In most of the articles, it is indicated that the data is processed in the cloud itself, and the end-users view all the information by connecting to the cloud. Cloud storage has been done on different platforms as it can be seen in Figure 27. In 24 of the reviewed articles, the authors specified that they store the monitored data in the cloud, but do not identify the utilized cloud platform (see Table 8). Most of the works in which the authors define the employed cloud platform, the utilized platform is Thingspeak (The MathWorks, Natick, MA, USA), with 14 papers. This platform is very intuitive and provides both free and paid options for storing, analyzing and displaying the data on different devices. Algorithms can be developed using MATLAB (The MathWorks, Natick, MA, USA) to generate alerts. However, there are proposals that use other cloud platforms such as FIWARE (FIWARE Foundation, Berlin, Germany) [75,149] and Dynamo DB (Amazon DynamoDB, Seattle, WA, USA) [117,159] with two papers each, and MongoDB (MongoDB Inc., New York, NY, USA) [28], Ubidots (Ubidots, Doral, FL, USA) [5], Amazon (Amazon, Seattle, WA, USA) [91], M2X (AT&T, Dallas, TX, USA) [200], NETPIE (Nexpie Co., Ltd., Bangkok, Thailand) [43], SAP (SAP SE, Walldorf, Germany) [201], InteGra (Integra Network Services, Milford, MA, USA) [194], Firebase (Firebase, San Francisco, CA, USA) [154], InfluxDB (InfluxData, San Francisco, CA, USA) [176]. These less-used platforms are either more expensive, provide fewer services or are less intuitive than Thingspeak. Table 4 lists the cloud platforms that have been used in the reviewed articles.

The rest of the articles where the authors identify the storage system utilize databases. As it can be seen in Figure 28, in 12 articles the information is stored in a database, but the name of the employed database is not specified. Other authors have identified the utilized databases. The most utilized one is MySQL [35,46,54,58,67,76,96,110,124,126,144,158] 12 papers, followed by SQL [69,119] with two papers. Other less utilized databases where SQLite (Hwaci. Hipp, Wyrick & Company, Inc., Charlotte, NC, USA) [52] and a NoSQL database with JSON [185] with one paper each. 

## 7. Discussion

In this section, current trends regarding IoT irrigation systems for agriculture are going to be presented.

### 7.1. Big Data Management and Analytics for Irrigation Optimization

IoT systems, in general, generate a great amount of data due to monitoring different parameters in real-time, and IoT irrigation systems generate big data as well. Therefore, it is necessary to provide mechanisms to manage and analyze big data. Ahad et al. [202] comment on the need for sustainable management of big data to avoid overusing natural resources. Using blockchain, powering IoT devices with solar energy, selecting useful data and discarding unnecessary and redundant data, employing clustering techniques to reduce the volume of the information, implementing time-efficient algorithms and the utilization of sustainable resources as alternatives to the components that are employed nowadays are some of the suggestions they provide.

Although the data gathered from the sensors is already a big source of information, the analysis of this data is paramount to optimize the IoT irrigation system according to the crop and the weather conditions. For this purpose, Tseng et al. [203] presented a big data analysis algorithm to aid farmers in the crop selection process. Their proposal satisfied the five Vs (volume, velocity, variety, veracity, and value) of big data. They performed a 3D correlation analysis that evaluated the irrigation cycle to identify the irrigation practices performed by the farmer. Then, they calculated the soil moisture content to detect irrigation and to evaluate if the farmer applied pesticides or fertilizers. This information was coupled with data on the weather conditions and other aspects of the soil to provide the farmer with the cultivation risks of each type of crop. Nonetheless, there are other technologies that are currently being utilized to analyze the big data produced by IoT irrigation systems for agriculture.

Artificial intelligence (AI) is currently the type of technology most companies are interested in for various purposes. In the case of irrigation, the use of AI is tied to the optimization of the available resources such as water, fertilizers or energy or to obtain information from the crops such as the presence of diseases or the correct growth of the plants, vegetables, and fruits. One of the techniques employed to analyze the data gathered from the sensors so as to perform decisions on the irrigation systems is fuzzy logic. Fuzzy logic has been employed to improve irrigation scheduling in greenhouses and control the drainage implementing it on the controller of the system [100]. The fuzzy rules were provided according to the water availability utilizing the QtFuzzyLite software ((FuzzyLite Limited, Wellington, New Zealand). Another fuzzy logic controller is employed to determine the coverage error for the resources so as to determine if the effective watering of the plants can be guaranteed [96]. The system presented in [136] utilized fuzzy logic to determine the duration of the irrigation process once the soil moisture values go below the 20% threshold. Light and temperature lead to the least accuracy in irrigation (40%) whereas any combination with the soil moisture values reached an accuracy from 62 to 81%. The authors conclude that the subsoil moisture sensor is the most important sensor to determine irrigation. The irrigation time was predicted in [162] as well, using the adaptive neuro fuzzy interference system (ANFIS) to predict the irrigation time. This fuzzy logic technique was implemented employing the MATLAB software. However, outdoor temperature and light are important to avoid the evaporation of the water. Another irrigation system utilizes iterative learning control (ILC) arithmetic to make decisions regarding the opening of the water valves [67]. 

Machine learning is another technique that is utilized in irrigation systems to perform predictions. The prediction algorithm in [120] for soil moisture prediction obtained an accuracy of 96%. Prediction techniques can also be utilized to estimate the amount of available water for irrigation such as the case of [119], where the authors employ Fisher’s LDA method for that purpose. Water usage is predicted as well in [50] employing regression algorithms. The regression algorithms consider the temperature of the environment and the water flow to produce a water consumption forecast that is then visualized by the user through a mobile app. 

Feedforward neural networks and the structural similarity index were employed in [204] to develop an optimized smart irrigation system. The authors compared gradient descent and variable learning rate gradient descent optimization techniques to predict soil moisture and they determined that the latter was the best one. Furthermore, a study on machine learning applied to agriculture so as to perform predictions on irrigation was performed in [205]. Different machine learning techniques were applied to the data on the soil and the environment collected from jojoba plantations. The authors conclude that the boosted tree classifier was the best classification model was and the gradient boosted regression trees was the best regression model. They had a 95% and 93% accuracy respectively. The authors suggest as well that different feature-sets would be better for different plots as some configurations achieved better results for different types of plots such as organic ones. Furthermore, the models considerably reduced their accuracy when deliberated soil drying was performed. Moreover, the authors state that the data on the weather and the saturation could be not considered without it substantially affecting the accuracy of the models.

### 7.2. Low-Cost Autonomous Sensors 

As we have seen in Section 2, the majority of the countries that dedicate more effort to researching IoT irrigation systems are countries where the farmers cannot afford commercial solutions. Therefore, most of the solutions for irrigation management are comprised of low-cost sensors and controllers so as to increase the scope of the benefits of utilizing smart irrigation systems. Apart from individual sensors, some shields integrate several sensors and connectors for irrigation and agriculture specific IoT systems such as the OpenGarden shield ((Libelium Comunicaciones Distribuidas S.L., Zaragoza, Spain) [136]. It can be connected to an Arduino UNO and it is comprised of an LDR, a DHT22 connector (SparkFun Electronics, Niwot, CO, USA), a soil moisture connector, a hydroponics connector, digital and analog pins, a DS18B20 connector (Adafruit Industries, New York, NY, USA) and an antenna connector. This type of solution allows the easy implementation of low-cost smart irrigation systems.

### 7.3. Sustainable Irrigation Systems

Most irrigation systems do not have access to the power grid, therefore there is a need for alternative means of powering up the devices that conform the system. Even irrigation systems with access to the power grid may only receive power during a constrained time period. In India, for example, the system presented in [29] determines when access to electricity is available to turn on the irrigation motors. This system was designed to help farmers when access to electricity happens at irregular hours or even at night, when farmers may not be present in the field. To address the need for alternative ways of powering the irrigation systems in agricultural fields that are too far to access the power grid, many systems incorporate a solar energy functionality, making the system more sustainable [41]. Furthermore, the use of solar energy reduces the energetic costs which is an advantage for irrigation systems intended for developing areas. Considering those reasons, a total of 22 of the surveyed papers chose to utilize solar energy to power their proposed irrigation system [4,38,41,46,50,56,63,66,69,88,90,91,92,96,99,101,140,145,171,178,181]. 

### 7.4. Frequency of the Data Acquisition

The frequency at which the parameters are monitored may be different for each proposal. Real-time monitoring [31,71] has been surging as the cost of the devices, whether they are processors, devices for data storage or sensors, has experienced a considerable reduction. Some proposals decide not to monitor the parameters in real-time so as to reduce energy consumption. Some authors have decided to take a measure every 10 s [58].

### 7.5. New Forms of Data Acquisition

The advances in technology as lead to new ways of obtaining the data from the sensors deployed in the fields. One of the new ways of gathering data from the sensor nodes is by utilizing drones [31]. Drones also allow obtaining new data that could not be obtained in other ways such as aerial images of the fields. 

Another new form of data acquisition is the use of robots that can incorporate both sensors and actuators to perform activities such as soil moisture sensing, weeding, spraying water or scaring the animals [65]. Robots can be utilized for the irrigation of precise areas due to their ability to travel to the desired location [38]. This robot is able to measure soil moisture and include proximity sensors to avoid collisions. The robot has a sprinkler system to irrigate the desired area. Furthermore, the wireless robot in [66] is comprised of both environmental and soil monitoring sensors and performs tasks such as water spraying, scaring animals and moving through the field. In order to improve the navigation of the robots for irrigation, the authors in [55] provided the robot with a coverage path planning (CPP) algorithm that has a map of the static elements and environmental data. The robot operating system (ROS) was utilized to develop the control system of the robot which is divided into three layers. The first layer reads the data from the sensors and controls the actuators, the second layer performs the communication among the elements of the system and the third layer performs the decision making and the path planning. This robot performs irrigation as a linear shower. Furthermore, these robots can be powered by solar panels [38,66] to provide them with more autonomy. 

### 7.6. Security in IoT Systems for Irrigation

Depending on the characteristics of the environment where the implementation of the IoT system is carried out, securing the system may become a challenging task as different types of threats must be considered. On the one hand, all the security issues of any IoT system can be applied to an IoT irrigation system for agriculture. Therefore, current works with their focus on securing IoT describes the security challenges that IoT irrigation systems may face. For example, organizations such as the Internet Engineering Task Force (IETF) study the different problems that affect this. Furthermore, authors such as Garcia-Morchon et al. [206] present in their work the following list of security threats and managing risks: vulnerable software/code, privacy threats, cloning of things, malicious substitution of things, eavesdropping attacks, man-in-the-middle attack, firmware attacks, extraction of private information, routing attack, elevation of privilege, denial-of-service (DoS) attacks. Regarding DoS attacks, Kamienski et al. [50] highly stressed the importance of protecting the irrigation system from DoS attacks or avoiding attackers from accessing the collected data or the system with the intention of damaging the crops.

In order to face the above threats, it is necessary to apply the appropriate security measures, but as new threats and attacks appear daily, [206] propose using methodologies such as analyzing the business impact of the attack, assessing the threats considering their impact and probability, its impact on the privacy of the collected information and the process of reporting and mitigating incidents.

However, on the other hand, when considering just the currently available studies on securing IoT irrigation systems specifically, it is possible to discern which aspects of IoT security are prioritized for this type of system. 

Jimenez et al. [207] identified the security threats specific to an intelligent wastewater purification system for irrigation. Apart from the previously mentioned security threats [206], the authors of [207] contemplate physical attacks that the IoT deployment may face. The IoT devices must be protected from the weather and possible animals that access the fields. However, strong weather conditions such as floods or animals such as wilds boars may result in the IoT device being damaged or lost. Furthermore, the IoT devices may also be damaged by the operators when they are performing their activities or be the aim of a person with malicious intentions. Another threat to the devices deployed on the field is the possibility of them being replaced to malicious nodes, providing the attacker with access to the network [29]. The deployed devices may also be susceptible to malicious code and false data injection, leading to wrong results and the malfunction of the system. Sleep deprivation attacks are aimed at depleting the battery of the devices. The deployed nodes are susceptible to booting attacks as well. Furthermore, attackers may also perform eavesdropping and interfere with the deployed devices. The concern of securing the physical devices deployed on the fields has led to many systems incorporating PIR sensors, scarecrows, and buzzers to detect intruders, either humans or animals, and to alert of their presence [59,208].

Privacy is another aspect to consider. The data managed by IoT irrigation systems may not need as much privacy as the data managed by other IoT applications such as those for health. In fact, Jimenez et al. [207] remark that it may not be necessary for the gathered data of some irrigation IoT systems to remain private, but the correct performance of the system must be ensured, which is avoiding DoS attacks. However, the owners of the fields may be implementing new techniques they would prefer to remain private or the water quality management system could be regarded as a critical infrastructure as the produce that results from the fields irrigated with that water would be consumed by people. Therefore, considering this aspect, Ahad et al. [202] and Barreto et al. [209] remark the need for user privacy, as the obtention of this information may lead to attacks to the owner or the personnel of the farm. Ahad et al. [202] consider data and device privacy as well, because of the need for guaranteeing the ownership of the data to avoid repudiation and ensuring data availability. Barreto et al. [209] remark on the burden that could be originated from the exposure of the GPS locational data captured by IoT devices as attackers may gain information on the location of the farm to perform physical attacks. Lastly, Ahad et al. [202] also comment on the need for a secure storage system for the information, with particular emphasis on distributed data storage systems. Attacks such as SQL injections can lead to the obtention of private information from storage systems [210].

As cloud computing often goes hand in hand with IoT systems, threats to cloud services may compromise the IoT irrigation system as well. Flooding attacks and cloud malware injection are some of the attacks that can be intentionally executed to compromise the data and the performance of the cloud [210].

Ransomware is another security threat that could affect irrigation systems for agriculture [209]. All the data regarding pesticides and the fertigation system could be encrypted until a ransom is paid. For this reason, having a back-up in a remote location is greatly advised.

Barreto et al. [209] also remark some security threats that are not usually commented on when discussing the security of IoT systems for agriculture. One of these threats is the damage that can be caused by social engineering as the users of the IoT irrigation system could provide login information to people posing as technicians or click on malicious links on their e-mails. The other threats are agroterrorism, cyber-espionage, mostly related to accessing intellectual property or confidential information regarding aspects such as genetically modified crops.

Blockchain is a new technique that is being applied to secure IoT systems [210], which allows secure data storage and communication. In agriculture, it is mostly utilized to secure the supply chain [211], although there are other proposals that use it to secure a greenhouse [212] or to secure the overall smart agriculture framework [213]. Regarding IoT irrigation systems for agriculture, blockchain has been applied in [214] to track and trace the information exchange of their proposed smart watering system.

### 7.7. Common Architecture Designs for IoT Irrigation Systems for Agriculture

In this subsection, an overview of the most frequent architectures for these systems is going to be discussed.

Multi-agent architectures are popular in IoT solutions for irrigation management [175,180]. These types of architectures make a distinction among the different elements they are comprised of. Typically, the distinction is made according to the layer of the architecture the elements are encompassed in. For example, nodes positioned higher in the hierarchy may act as a broker for those lower in the hierarchy [175]. 

Most architectures are divided into layers or functional blocks that represent the main actions to be carried out [215]. These blocks or layers can be considered as generic and are utilized in most of the available architecture designs for IoT systems intended for irrigation management. The main elements of these architectures are devices, communications, services, management, applications, and security. 

IoT systems are comprised of devices located in a certain environment to perform multiple activities such as detection, monitoring, control, and action. The devices must have interfaces that allow their connection with other devices to transmit the necessary information. The data obtained through different sensors will generally be treated and their results will be applied to different actuators. Then, it is necessary to transmit the monitored data and response actions between the devices. Communication protocols are used for this task. In most cases, different communication protocols are used working together on the same IoT system. In order to perform activities such as device discovery, device control or the analysis of the collected data, the use of services may be necessary as well. The applications allow the interaction of the user with the system. From the applications, the user will be able to visualize the information, both obtained by monitoring and that extracted from the data once they have been processed. On many occasions, the user can execute the actions that he considers appropriate to the situation presented by the data, the actions can also be carried out automatically. Lastly, providing security to the system may also be considered. 

Traditionally, the IoT architecture has been considered to be divided into three layers, which are the perception, network, and application layer. Later on, an intermediate layer placed between the network and application layers has been introduced in multiple studies. This layer, called service layer, is employed to store and process data using cloud and fog computing. For the last few years, authors such as Ferrández-Pastor [216] presented a new architecture proposal, based on four layers: things, edge, communication, and cloud. In these latest architectural proposals, the authors use the edge layer to locate critical applications and perform basic control processes. Also, as indicated by [216], cloud (internet/intranet) can provide Web services, data storage, HMI interfaces or analytic applications. Figure 29 shows an image where one can see the architecture models.

In the case of the reviewed IoT systems for irrigation, both 3-layered [46,83,118,217,218] and 5-layered architectures [180] are available. Commonly, the lower layer is comprised of the sensor nodes and the actuators. The middle layer is comprised of a gateway and contemplates data transmission. Lastly, the third layer is usually tasked with performing data storage and analysis. Typical third layers are conformed of cloud services, databases or applications.

A novel take on IoT deployments for precision agriculture is being considered on the Internet of Underground Things [219]. The authors define the functionalities as in-situ sensing, wireless communication in underground environments and the connection between the elements of the architecture such as sensors, machinery or the cloud. For the case of IoUT, the sensors are deployed underground. The authors performed a study on wireless communication among above ground and underground devices. The path loss link between above ground and underground devices reached −80 dBm for 50 m. The distance between underground devices reached approximately 10 m for −80 dBm. The authors also comment on the effects of soil moisture on path loss.

### 7.8. Recommendations for the Implementation of a Smart Irrigation System for Agriculture

In this subsection, we present an architecture proposal for an IoT irrigation system for irrigation. In order to ensure the correct performance of the IoT irrigation system for precision agriculture, the architecture should provide interoperability, scalability, security, availability, and resiliency.

After having carried out an exhaustive study of works presented by other researchers, our architecture proposal is divided into four layers that we define as devices, communication, services, and application, as shown in Figure 30. Furthermore, the aspects related to management and security should be treated jointly between the communication and services layers.

The first layer is the Device layer, where all the devices that will perform the functions of detection, monitoring, control, and action are located. We would have four types of nodes. The water monitoring node would monitor water quality to determine if the water is apt for the irrigation of the crops. The soil monitoring node would monitor soil moisture, soil temperature and other parameters that aid in the irrigation schedule decision process. The weather monitoring node would measure air temperature and humidity, precipitation, luminosity, radiation, and wind parameters so as to aid in the decision making process. Lastly, the actuator nodes would perform the actions resulting from the decision-making process.

The second layer is the communication layer, comprised of three blocks. The Hop-to-Hop communication block allows defining data link layer technologies and to transmit the frames that contain the data of the device layer. From this block the frames will be sent to the network communication block, to reach the remote locations. The routing function can be assumed in this block in mesh networks, such as 802.15.4 networks. The end-to-end communication block is responsible for delivering the capabilities of the transport and application layers of the TCP/IP model when communication crosses different network environments. Finally, the network communication block is responsible for the communication (routing) between the networks, of the hop-to-hop at end-to-end blocks, using the address, for example, IPv4 and IPv6, and the ID resolution. In addition, it will be responsible for managing the QoS.

The next layer is the services layer, which is comprised of three blocks. The services block includes the IoT services and operability for their discovery and search. The organization block performs the assignment of the services according to the needs of the users or the available resources. At last, in business environments that relate to IoT, the modeling and execution of service block will be invoked from the execution of the applications.

There are two blocks, management and security, that act on both the previously described communication and services layers. The management block is based on the fault, configuration, accounting, performance, security (FCAPS) model and framework. This is a model of the ISO Telecommunications Management Network [220].

The security block guarantees both the security and privacy of the systems and is comprised of four blocks. The authentication block is responsible for the authentication of users and services. The authorization block manages access control policies. Furthermore, access control decisions will be made based on access control policies. The key exchange & management block is used to achieve secure communications between peers. Finally, the trust & reputation block is responsible for scoring the user and calculating the level of trust of the service.

The last layer is the application layer. It allows users to interact with the IoT system. From this layer, users can receive alarms, visualize the gathered data in real-time or activate the actuators or actions that have not been programmed automatically.

### 7.9. Future Challenges of IoT Irrigation Systems

The Internet of Underground Things (IoUT) is a new view of the IoT [221]. It consists of deploying both underground and above ground IoT devices that communicate among themselves with underground-underground, underground-above ground, and above ground-above ground communication. It is especially useful and applicable to irrigation and precision agriculture IoT systems as the devices would not be impeding the work of machines and farmers and it would reduce the amount of physical damage the nodes deployed on the fields may receive. Although WiFi surprisingly allows underground-above ground communication in short distances and depths above 30 cm [222], it would be necessary to study the performance of other existing protocols or even the creation of a new protocol that employs lower frequency bands to transmit the information through the soil medium. 

The use of LoRa is increasing for irrigation and precision agriculture applications due to its long-range, which allows wireless communication to remote fields. However, the advances in 5G may lead to a decrease in the interest in LoRa technology. On the one hand, if the designed irrigation system produces lower amounts of data, as these types of systems present low variability in the data, LoRa presents itself as a very good solution. However, on the other hand, for IoT systems that require the transmission of large amounts of data, 5G would solve the problem of the limited amount of data that LoRa can transmit. Considering these aspects, a possible future is the use of LoRa and 5G hybrid wireless networks to attempt in satisfying those needs [223].

There are diverse opinions on the existence and effects of climate change on earth. However, there is no doubt that many governments, companies and the citizens themselves are increasing their awareness regarding sustainable consumption. This not only affects their view on the energy consumption of IoT systems for irrigation but also, the origin of the components of the devices and the impact of IoT devices on the environment and the fauna of the area where they are deployed. Therefore, the reduction in energy consumption and the use of alternative powering methods will continue to be a research trend. Furthermore, it may be a challenge to find new materials that are weather-resistant and do not increase severely the cost of the devices, but the use of recyclable materials in some of the elements of the IoT devices is to be expected [203].

As it has further been commented on Section 7.1, blockchain and AI will be incorporated into most IoT services [210], including those aimed at irrigation and precision agriculture. These technologies provide not only increased security but also optimization to the management of the IoT systems. This will lead to an increased understanding of the crops and a faster way to characterize genetically modified crops and the effects of new fertilizers and pesticides. Furthermore, it will aid in optimizing and reducing water consumption and improving the new mechanisms that are being proposed to determine if treated wastewater can be used for irrigation and which crops can be irrigated with wastewater depending on its composition. 

Lastly, farmers usually have a very small profit and thus, these IoT systems may not be affordable to them. Therefore, the cost of IoT devices and the overall system implementation should have a decreasing tendency for these types of systems to have a commercial future.

## 8. Conclusions and Future Work

Water management is paramount in countries with water scarcity. This also affects agriculture, as a large amount of water is dedicated to this use. The rising concerns about global warming have led to the consideration of creating water management measures to ensure the availability of water for food production and consumption. Thus, the researches on water usage reduction for irrigation have increased over the years. In this paper, we have provided an overview of the actual state of the art regarding IoT irrigation systems for agriculture. We have identified the most monitored parameters to characterize water quality for irrigation, soil and weather conditions. We have also identified the most utilized nodes to implement IoT and WSN systems for the irrigation of crops and the most popular wireless technologies. Furthermore, the current trends in the implementation of IoT systems for crop management and irrigation have been discussed as well. We have provided a 4-layer architecture proposal as well for the management of crop irrigation. 

As future work, we are developing a smart irrigation system that evaluates water quality prior to irrigation based on the proposed architecture.

## Figures and Tables

**Figure 1 sensors-20-01042-f001:**
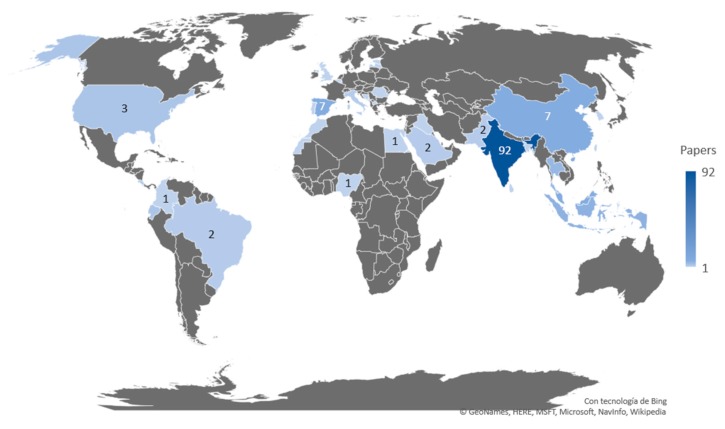
Number of published papers presenting IoT systems for irrigation per country.

**Figure 2 sensors-20-01042-f002:**
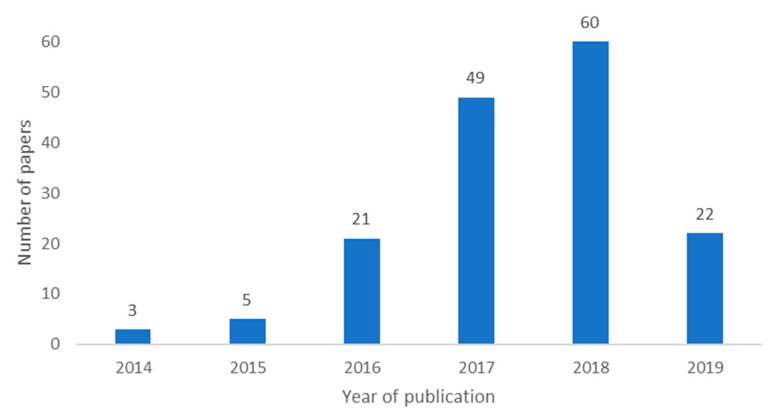
Number of published papers presenting IoT systems for irrigation per year.

**Figure 3 sensors-20-01042-f003:**
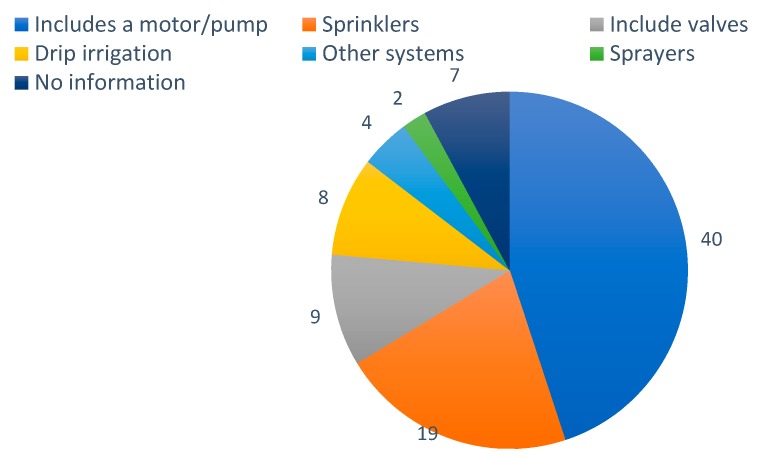
Number of papers that propose different irrigation systems.

**Figure 4 sensors-20-01042-f004:**
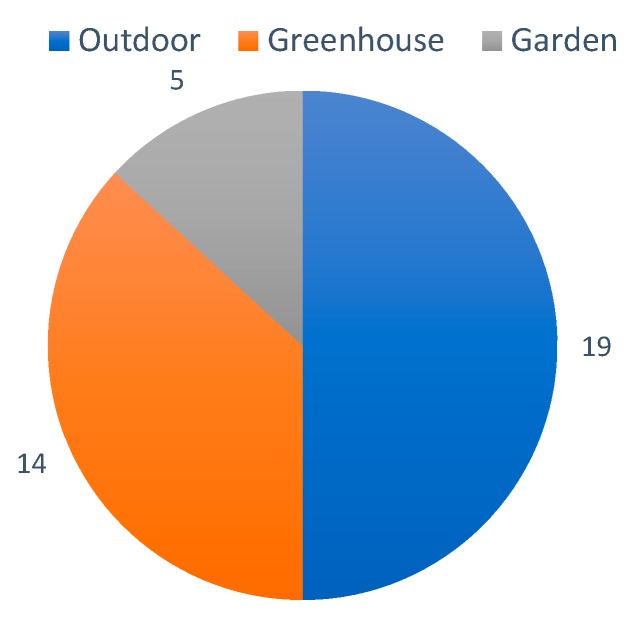
Number of papers that proposed irrigation systems for different sorts of agriculture.

**Figure 5 sensors-20-01042-f005:**
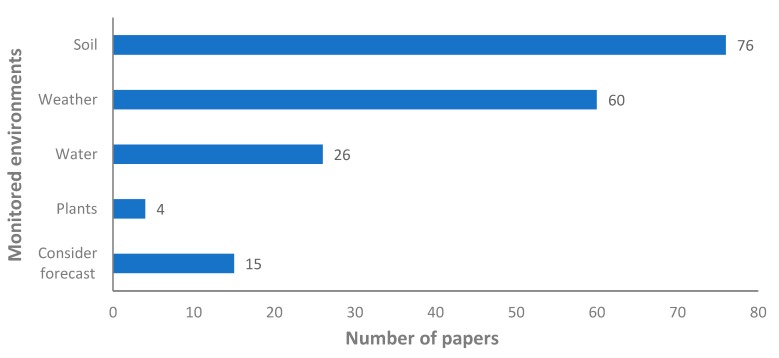
Monitored environments in papers that propose an irrigation system.

**Figure 6 sensors-20-01042-f006:**
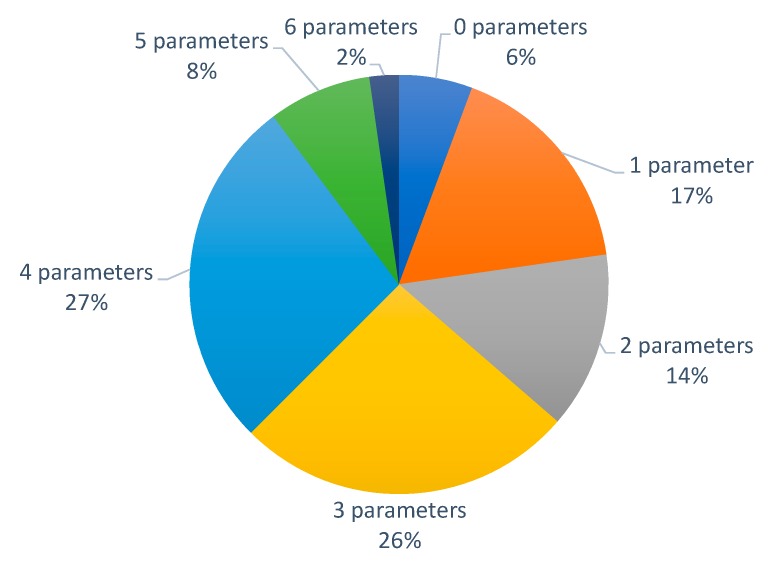
Number of monitored parameters in papers that propose an irrigation system.

**Figure 7 sensors-20-01042-f007:**
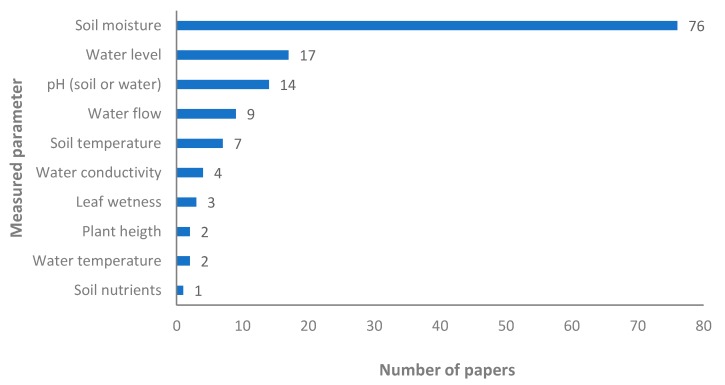
Monitored parameters from the soil, water, and plants in papers that propose an irrigation system.

**Figure 8 sensors-20-01042-f008:**
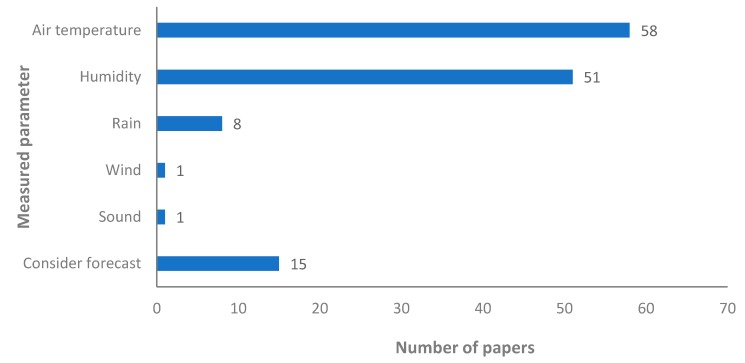
Monitored atmospheric parameters in papers that propose an irrigation system.

**Figure 9 sensors-20-01042-f009:**
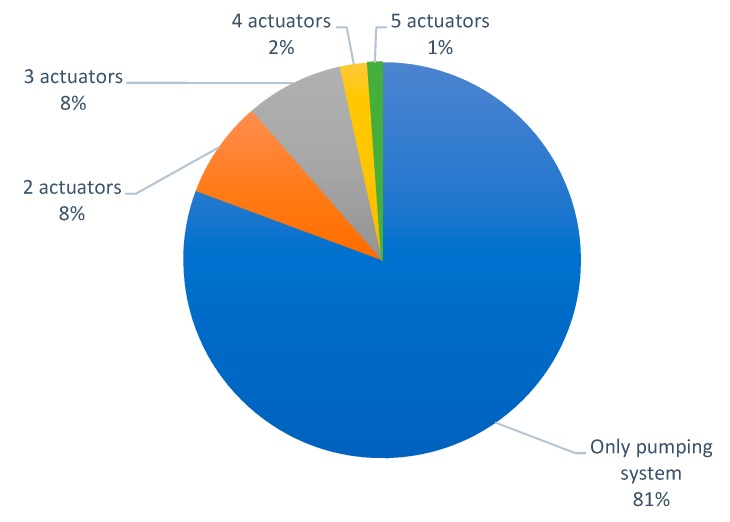
Number of included actuators in papers that propose an irrigation system.

**Figure 10 sensors-20-01042-f010:**
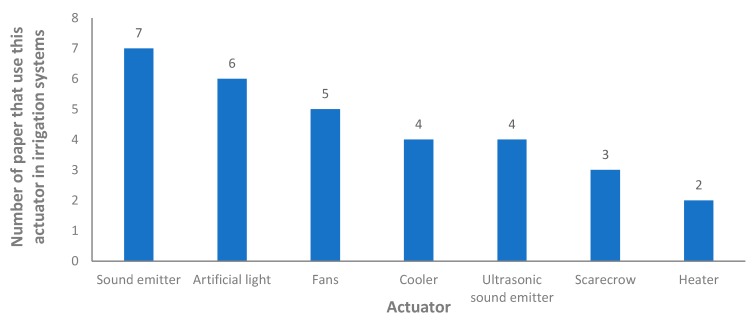
Used actuators in papers that propose an irrigation system.

**Figure 11 sensors-20-01042-f011:**
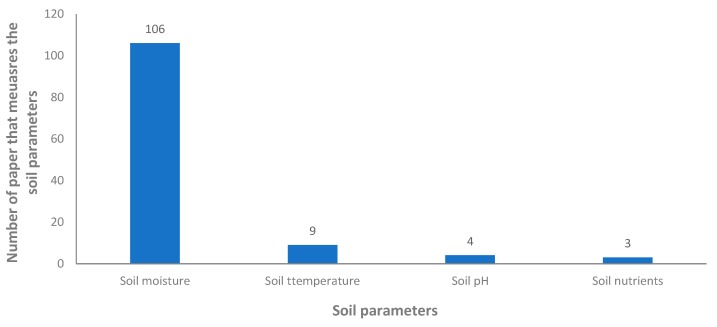
Monitored soil parameters in all the evaluated papers.

**Figure 12 sensors-20-01042-f012:**
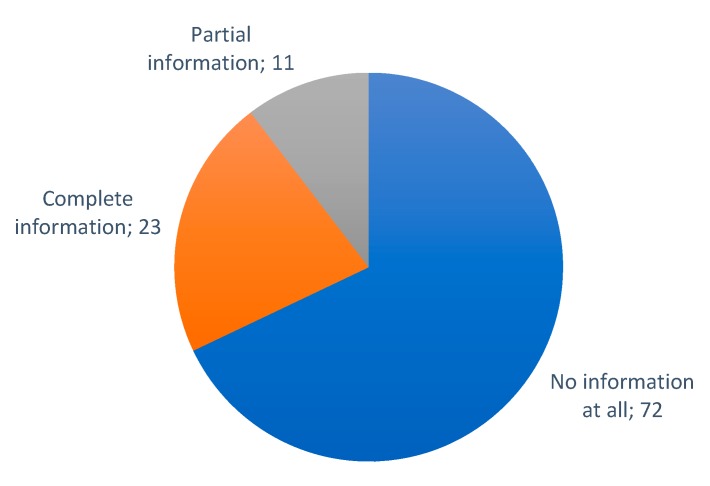
Information about the used moisture sensors in all the evaluated papers.

**Figure 13 sensors-20-01042-f013:**
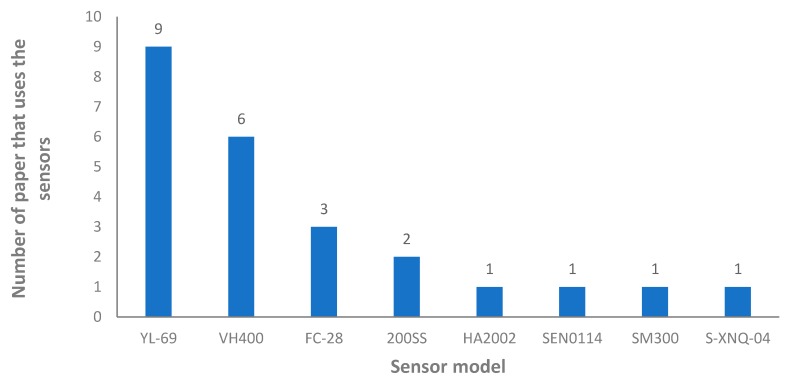
Number of papers that used different models of soil moisture sensors.

**Figure 14 sensors-20-01042-f014:**
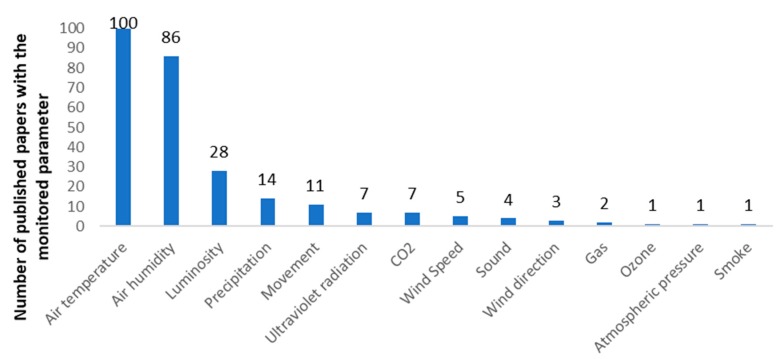
Most monitored weather parameters.

**Figure 15 sensors-20-01042-f015:**
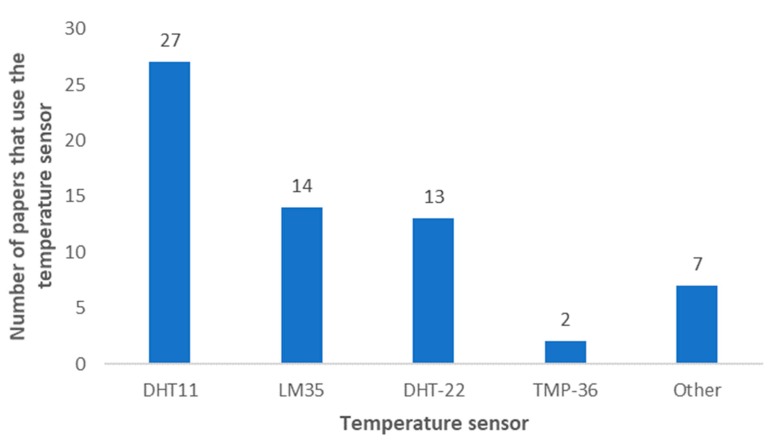
Most utilized temperature sensors.

**Figure 16 sensors-20-01042-f016:**
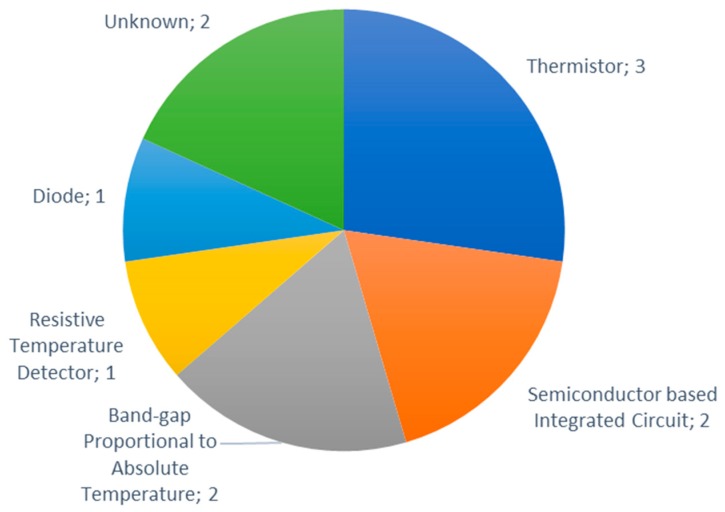
Most utilized temperature sensors by type.

**Figure 17 sensors-20-01042-f017:**
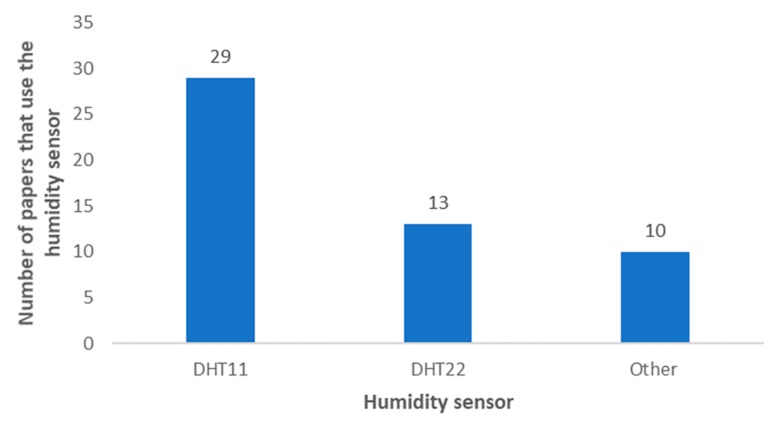
Most utilized humidity sensors.

**Figure 18 sensors-20-01042-f018:**
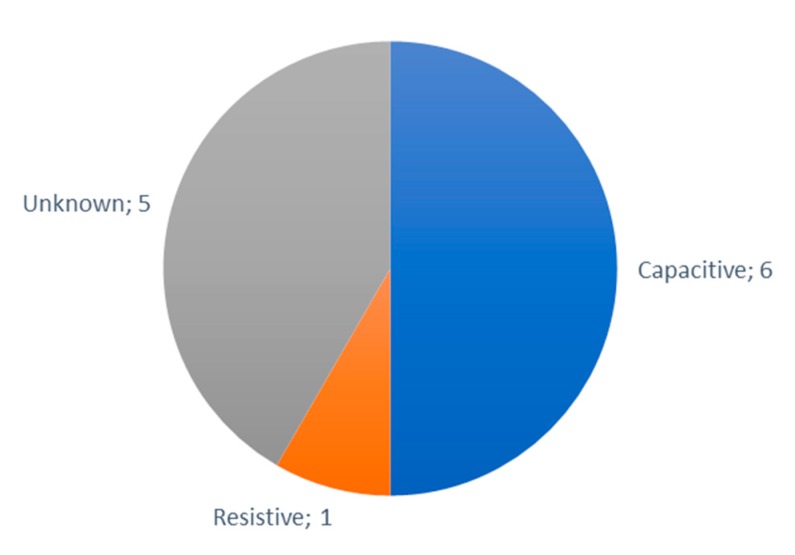
Most utilized humidity sensors by type.

**Figure 19 sensors-20-01042-f019:**
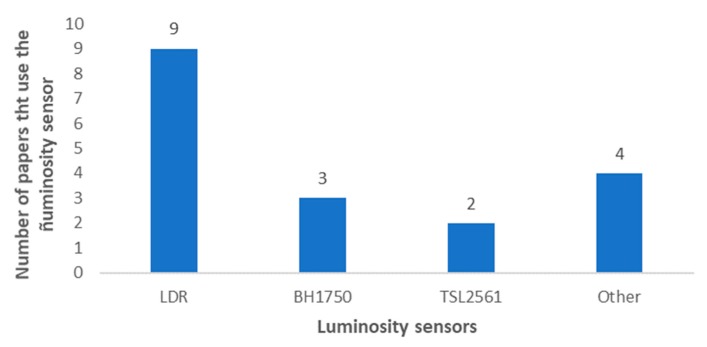
Most utilized luminosity sensors.

**Figure 20 sensors-20-01042-f020:**
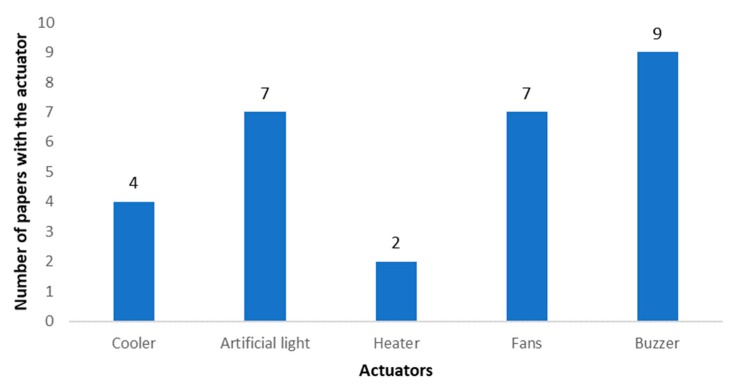
Most utilized actuators.

**Figure 21 sensors-20-01042-f021:**
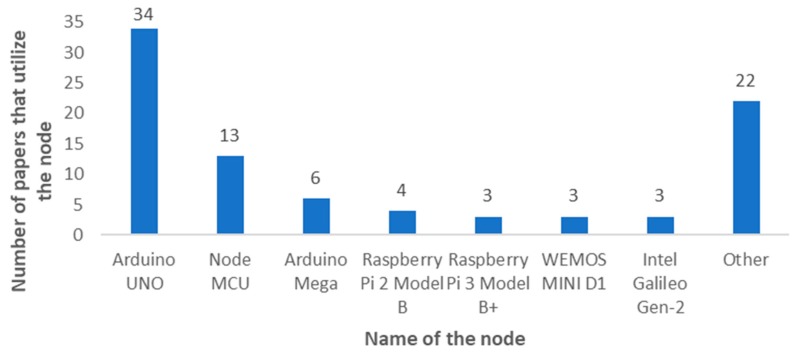
Most utilized nodes to implement IoT irrigation systems.

**Figure 22 sensors-20-01042-f022:**
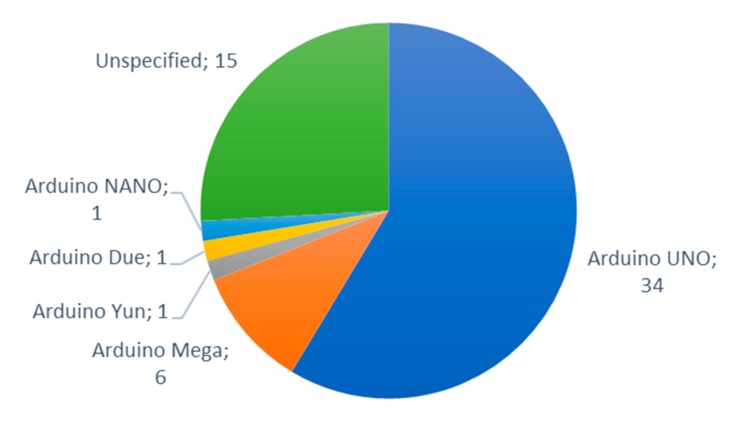
Arduino nodes utilized to implement IoT irrigation systems.

**Figure 23 sensors-20-01042-f023:**
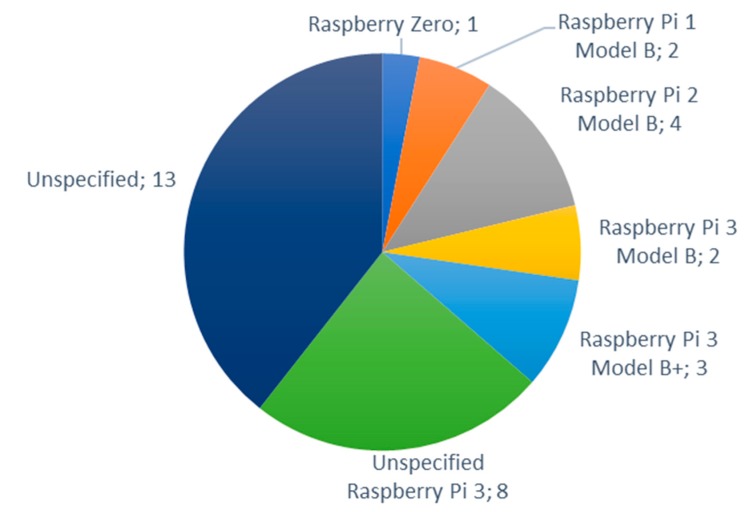
Raspberry Pi nodes utilized to implement IoT irrigation systems.

**Figure 24 sensors-20-01042-f024:**
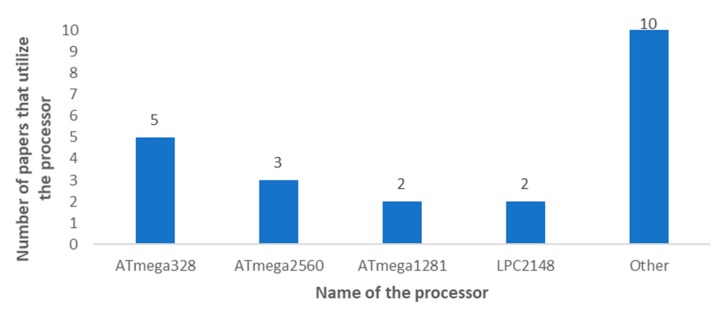
Most utilized processors to implement IoT nodes for irrigation systems.

**Figure 25 sensors-20-01042-f025:**
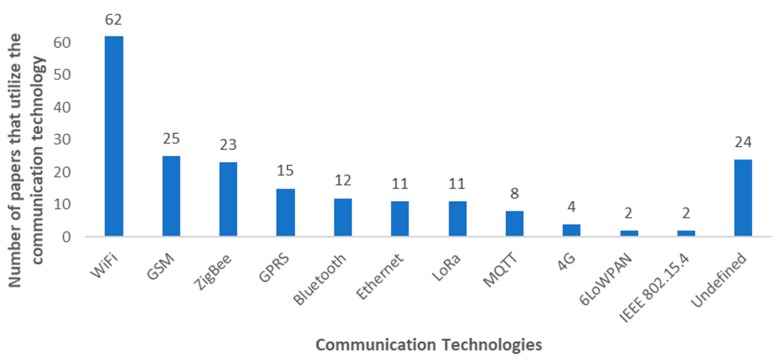
Communication technologies employed to implement IoT irrigation systems.

**Figure 26 sensors-20-01042-f026:**
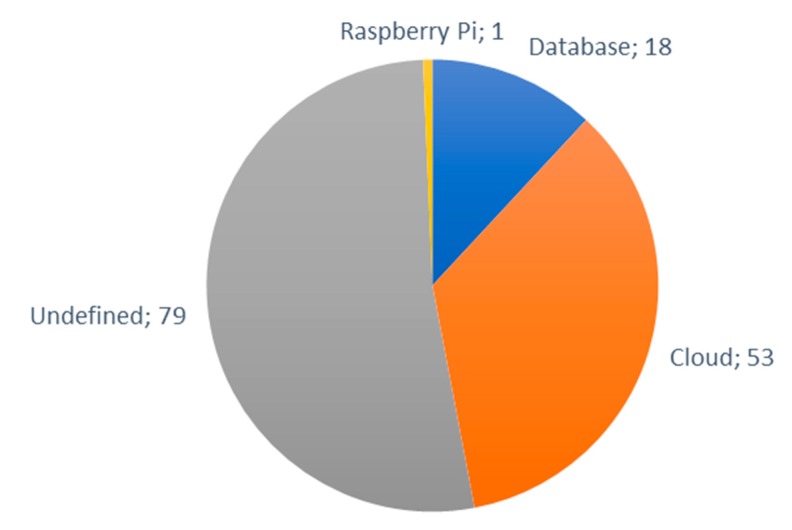
Number of papers per storage system.

**Figure 27 sensors-20-01042-f027:**
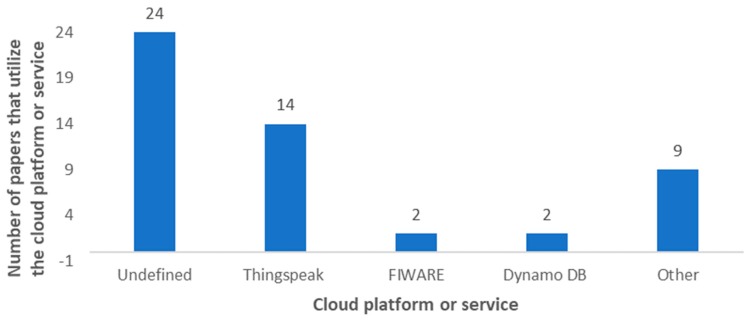
Number of papers that employ each clouds service or platform.

**Figure 28 sensors-20-01042-f028:**
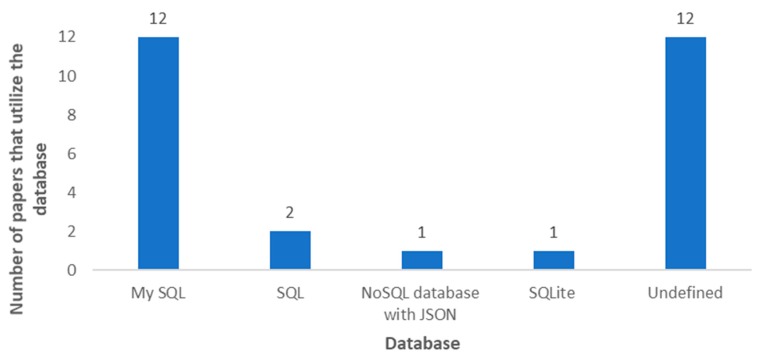
Number of employed Databases to store data of IoT nodes for irrigation systems.

**Figure 29 sensors-20-01042-f029:**
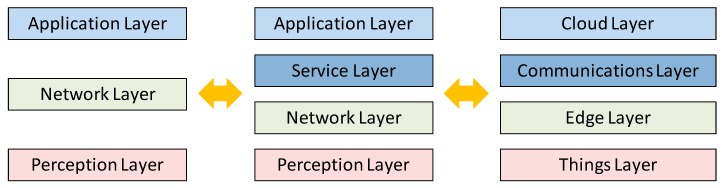
Evolution of the layered model in IoT architecture.

**Figure 30 sensors-20-01042-f030:**
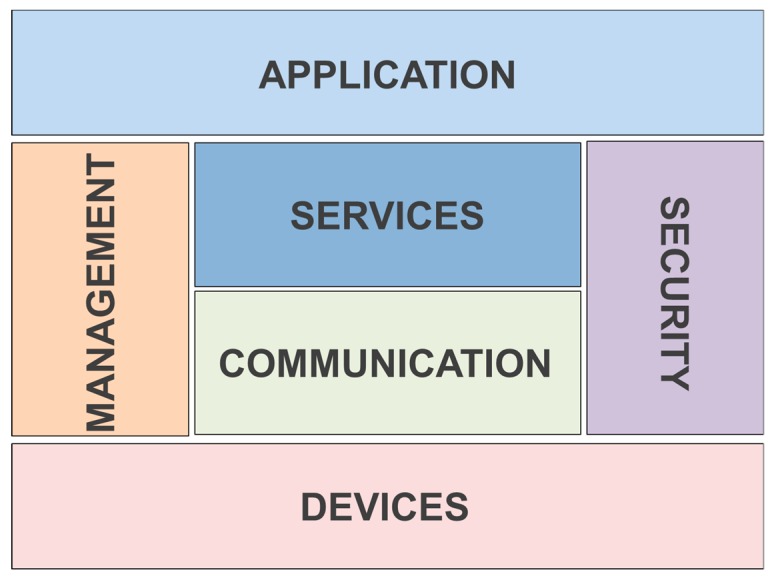
Architecture proposal for an IoT irrigation system for agriculture.

**Table 1 sensors-20-01042-t001:** Papers that measure the most monitored weather parameters.

Parameter	References
Air temperature	[28,32,34,35,45,46,47,48,49,50,51,52,56,57,58,59,61,62,63,64,65,66,67,68,71,72,73,74,75,76,80,81,82,83,84,85,86,87,88,89,91,93,94,96,97,98,99,100,102,103,111,114,115,117,118,119,120,121,122,124,125,126,127,128,129,130,131,132,133,134,135,136,137,138,139,140,141,142,143,144,145,146,147,148,149,150,151,152,153,154,155,156,157,158,159,160,161,162]
Relative humidity	[14,28,34,35,42,45,46,47,48,49,51,57,58,59,61,62,63,64,65,66,67,68,71,72,73,74,75,76,81,84,85,86,87,88,89,91,93,94,95,96,97,98,99,100,102,103,111,114,115,117,118,119,120,121,122,125,126,127,130,131,133,135,137,138,139,140,142,143,144,145,146,148,150,153,154,155,156,157,158,159,160,162,163,164,165]
Precipitation	[30,46,48,50,61,67,76,95,114,119,137,143,157]
Luminosity	[28,45,48,62,63,65,66,67,71,72,76,91,99,112,118,119,125,126,129,131,136,140,144,145,154,156,157,159]

**Table 2 sensors-20-01042-t002:** Temperature monitoring sensors.

Sensor	Temperature Range		Accuracy	Reference
	Min	Max		
DHT11	0 °C	50 °C	±2 °C	[28,59,61,62,64,71,87,93,94,101,103,115,117,118,119,121,130,131,135,139,143,144,148,154,159,160,162]
DHT22	−40 °C	125 °C	±0.5 °C	[76,88,95,96,114,120,126,144,146,151,155,156,158]
LM35	−55 °C	150 °C	-	[48,49,53,56,65,80,82,84,91,98,103,129,132,165]

**Table 3 sensors-20-01042-t003:** Relative humidity monitoring sensors.

Sensor	Humidity Range		Accuracy	Reference
	Min	Max		
DHT11	20%	90%	±5%	[28,59,61,62,64,71,87,93,94,101,103,115,117,118,119,121,130,131,135,139,143,145,148,154,156,159,160]
DHT22	0%	100%	±5%	[76,88,95,96,114,120,126,144,146,151,155,156,158]

**Table 4 sensors-20-01042-t004:** Most popular nodes for IoT irrigation systems.

Node	Reference
WEMOS MINI D1	[71,87,89]
Node MCU	[41,43,86,97,113,114,127,153,155,158,167,168,169]
Arduino Mega	[52,55,56,72,170,171]
Arduino UNO	[34,36,40,44,49,54,55,61,62,64,69,76,84,91,111,112,117,119,121,127,129,139,143,144,151,153,154,156,160,166,168,172,173,174]
Raspberry Pi 2 Model B	[66,69,165,175]
Raspberry Pi 3 Model B+	[58,112,176]
Intel Galileo Gen-2	[30,51,77]

**Table 5 sensors-20-01042-t005:** Main technologies used in IoT for irrigation systems.

Group	Technology	Frequency Bands	Max. Data Rate
Cellular	3G	380.2–389.8 MHz, 390.2–399.8 MHz, 410.2–419.8 MHz, 420.2–429.8 MHz, 450.6–457.6 MHz, 460.6–467.6 MHz, 479.0–486.0 MHz, 489.0–496.0 MHz, 698.2–716.2 MHz, 728.2–746.2 MHz, 777.2–792.2 MHz, 747.2–762.2 MHz, 806.2–821.2 MHz, 851.2–866.2 MHz, 824.2–848.8 MHz, 869.2–893.8 MHz, 890.0–915.0 MHz, 935.0–960.0 MHz, 880.0–915.0 MHz, 925.0–960.0 MHz, 876.0–915.0 MHz, 921.0–960.0 MHz, 870.4–876.0 MHz, 915.4–921.0 MHz, 1710.2–1784.8 MHz, 1805.2–1879.8 MHz, 1850.2–1909.8 MHz, 1930.2–1989.8 MHz	Upload 7.2 Mbps Download 2 Mbps
	4G/LTE	452.5–457.5 MHz, 462.5–467.5 MHz, 703–748 MHz, 703–803 MHz, 704–716 MHz, 717–728 MHz, 734–746 MHz, 738–758 MHz, 758–803 MHz, 791–821 MHz, 807–824 MHz, 814–849 MHz, 815–830 MHz, 824–849 MHz, 830–845 MHz, 832–862 MHz, 852–869 MHz, 859–894 MHz, 860–875 MHz, 869–894 MHz, 875–890 MHz, 880–915 MHz, 925–960 MHz, 1427.9–1447.9 MHz, 1475.9–1495.9 MHz, 1447–1467 MHz, 1447.9–1462.9 MHz, 1495.9–1510.9 MHz, 1525–1559 MHz, 1626.5–1660.5 MHz, 1710–1770 MHz, 1710–1755 MHz, 1710–1780 MHz, 1710–1785 MHz, 1749.9–1784.9 MHz, 1805–1880 MHz, 1844.9–1879.9 MHz, 1850–1910 MHz, 1850–1915 MHz, 1880–1920 MHz, 1910–1930 MHz, 1930–1990 MHz, 1930–1995 MHz, 1920–1980 MHz, 1920–2010 MHz, 2000–2020 MHz, 2010–2025 MHz, 2110–2155 MHz, 2110–2170 MHz, 2180–2200 MHz, 2300–2400 MHz, 2305–2315 MHz, 2350–2360 MHz, 2496–2690 MHz, 2500–2570 MHz, 2620–2690 MHz, 3400–3600 MHz, 3410–3490 MHz, 3510–3590 MHz, 3600–3800 MHz, 5150–5925 MHz	Upload 150 Mbps Download 50 Mbps
	5G	Frequency Range 1: 663–698 MHz, 617–652 MHz, 699–716 MHz, 703–748 MHz, 717–728 MHz, 729–746 MHz, 758–803 MHz,788–798 MHz, 758–768 MHz, 791–821 MHz, 815–830 MHz, 824–849 MHz, 832–862 MHz, 860–875 MHz, 869–894 MHz, 880–915 MHz, 925–960 MHz, 1427–1432 MHz, 1427–1470 MHz, 1432–1517 MHz, 1475–1518 MHz, 1695–1710 MHz, 1710–1780 MHz, 1710–1785 MHz, 1805–1880 MHz, 1850–1910 MHz, 1850–1915 MHz, 1880–1920 MHz, 1920–1980 MHz, 1920–2010 MHz, 1930–1990 MHz, 1930–1995 MHz,1995–2020 MHz, 2010–2025 MHz, 2110–2170 MHz, 2110–2200 MHz, 2300–2400 MHz, 2305–2315 MHz, 2350–2360 MHz, 2496–2690 MHz, 2500–2570 MHz, 2570–2620 MHz, 2620–2690 MHz, 3300–4200 MHz, 3300–3800 MHz, 3550–3700 MHz, 4400–5000 MHz Frequency Range 2: 26.50–29.50 GHz, 24.25–27.50 GHz, 37.00–40.00 GHz, 27.50–28.35 GHz	
Wireless Personal Area Networks	IEEE 802.15.1—Bluetooth	2400–2483.5 MHz	Up to 3 Mbps
	BLE (Bluetooth Low-Energy)	2.400–2.4835 GHz	Up to 2 Mbps
RFID	Radio Frequency Identification (RFID)	Low Frequency: 125 or 134.2 kHz High Frequency: 13.56 MHz Ultra High Frequency: 868–956 MHz Microwaves: 2.45	UHF: Up to 640 kbps
Mesh Protocols	Zigbee	868–868.6 MHz, 902–9286 MHz, 2400 MHz	Up to 250 kbps
	Z-Wave	865.2 MHz, 869 MHz, 868.4 MHz, 868.40 MHz, 868.42 MHz, 869.85 MHz, 908.4 MHz, 908.42 MHz, 916 MHz, 919.8 MHz, 921.4 MHz, 919–923 MHz, 920–923 MHz, 920–925 MHz, 922–926 MHz	Up to 100 kbps
	Thread	Global: 2400–2500 MHz America, Australia: 902–928 MHz Europe: 868–868.6 MHz,	Up to 250 kbps
WiFi	IEEE 802.11a	5725–5875 MHz	Up to 54 Mbps
	IEEE 802.11b	2400–2500 MHz	Up to 11 Mbps
	IEEE 802.11g	2400–2500 MHz	Up to 54 Mbps
	IEEE 802.11n	2400–2500 MHz, 5725–5875 MHz	Up to 600 Mbps
	IEEE 802.1ac	2400–2500 MHz, 5725–5875 MHz	Up to 3.46 Gbps
	IEEE 802.11ah	AH All Countries: 2400–2500 MHz, 5725–5875 MHz AH Europe: 863–868 MHz AH USA: 902–928 MHz AH China:755–787 MHz AH Japan: 916.5–927.5 MHz AH Korea: 917.5–923.5 MHz AH Singapore: 866–869 MHz, 920–925 MHz	Up to 40 Mbps
Low-Power Wide Area Network (LPWAN)	NarrowBand IoT (NB-IoT)	Global: 1950 MHz, 2140 MHz, 1747.5 MHz, 1842.5 MHz, 897.5 MHz, 942.5 MHz, 455 MHz, 465 MHz	200 kbps
		EMEA: 847 MHz, 806 MHz, 453.5 MHz, 463.5 MHz	
		EU: 725.5 MHz, 780.5 MHz	
		North America Region: 1880 MHz, 1960 MHz, 1732.5 MHz, 2132.5 MHz, 836.5 MHz, 881.5 MHz, 707.5 MHz, 737.5 MHz, 782 MHz, 751 MHz, 793 MHz, 763 MHz, 710 MHz, 740 MHz, 1882.5 MHz, 1962.5 MHz, 831.5 MHz, 876.5 MHz, 1745 MHz, 2155 MHz, 1702.5 MHz, 2007.5 MHz, 680.5 MHz, 634.5 MHz, 1448.5 MHz, 1496.5 MHz, 707 MHz, 737 MHz	
		APAC: 725.5 MHz, 780.5 MHz, 452.5 MHz, 462.5 MHz	
		Japan: 1437.9 MHz, 1485.9 MHz, 822.5 MHz, 867.5 MHz, 837.5 MHz, 882.5 MHz, 1455.4 MHz, 1503.4 MHz	
	Long Term Evolution—Machine Type Communication (LTE-M)	Global: 1950 MHz, 2140 MHz, 1747.5 MHz, 1842.5 MHz, 897.5 MHz, 942.5 MHz, 455 MHz, 465 MHz	Upload peak rate of 5 Mbps Download peak rate of 10 Mbps
		EMEA: 2535 MHz, 2655 MHz, 847 MHz, 806 MHz, 453.5 MHz, 463.5 MHz	
		EU: 725.5 MHz, 780.5 MHz	
		North America Region: 1880 MHz, 1960 MHz, 1732.5 MHz, 2132.5 MHz, 836.5 MHz, 881.5 MHz, 707.5 MHz, 737.5 MHz, 782 MHz, 751 MHz, 793 MHz, 763 MHz, 1882.5 MHz, 1962.5 MHz, 831.5 MHz, 876.5 MHz, 815.5 MHz, 860.5 MHz, 1745 MHz, 2155 MHz, 680.5 MHz, 634.5 MHz, 1448.5 MHz, 1496.5 MHz, 707 MHz, 737 MHz	
		APAC: 452.5 MHz, 462.5 MHz, 725.5 MHz, 780.5 MHz	
		China: TDD, 1900 MHz, TDD, 2350 MHz	
		Japan: 1437.9 MHz, 1485.9 MHz, 822.5 MHz, 867.5 MHz, 837.5 MHz, 882.5 MHz, 1455.4 MHz, 1503.4 MHz	
	Extended coverage GSM (EC-GSM)	Global: 385 MHz, 395 MHz, 415 MHz, 425 MHz, 454 MHz, 464 MHz, 482.4 MHz, 492,4 MHz, 707.2 MHz, 737.2 MHz, 785.2 MHz, 755.2 MHz, 813.7 MHz, 858.7 MHz	Downlink Peak Data Rate: 70 kbps (GSMK), 240 kbps (8PSK) Uplink Peak Data Rate: 70 kbps (GSMK), 240 kbps (8PSK)
		APAC & EMEA: 897 MHz, 942 MHz	
		Caribbean, LATAM & North America Region: 836.5 MHz, 881.5 MHz, 1880 MHz, 1960 MHz	
		ITU Region 1 & 3: 902.5, 947.5 MHz, 895.5 MHz, 940.5 MHz, 1747.5 MHz, 1842.5 MHz	
	Sigfox	RC1: 868.130 MHz, 869.525 MHz RC2: 902.200 MHz, 905.200 MHz RC3: 923.200 MHz, 922.200 MHz RC4: 920.800 MHz, 922.300 MHz RC5: 923.300 MHz, 922.300 MHz RC6: 865.200 MHz, 866.200 MHz	100 or 600 bps
	LoRa—Low Power Wide Area Network (LoRaWAN)	Europe: 870 MHz, 863 MHz, 434 MHz, 433 MHz India: 867 MHz, 865 MHz Region 2—America, Greenland, Eastern Pacific Islands: 928 MHz, 902 MHz Australia, New Zealand: 928 MHz, 915 MHz China: 510 MHz, 470 MHz, 787 MHz, 779 MHz Hong Kong: 925 MHz, 920 MHz Taiwan: 928 MHz, 922 MHz South Korea: 923 MHz, 920 MHz Japan: 928 MHz, 920 MHz Singapore, Thailand, Vietnam: 925 MHz, 920 MHz Brunei, Cambodia, Indonesia, Laos: 925 MHz, 923 MHz	0.3 to 50 kbps
	MIOTY	868 MHz	

**Table 6 sensors-20-01042-t006:** Most utilized communication technologies in IoT crop irrigation systems.

Technology	References
Ethernet	[52,62,72,99,117,131,138,139,146,152,164]
GSM	[35,36,44,48,51,57,61,64,72,73,79,85,90,91,92,103,112,129,153,157,185,186,187,188,189]
Wi-Fi	[28,34,41,43,46,47,50,51,52,53,56,59,60,64,65,66,69,70,71,77,82,84,88,94,95,111,112,116,117,119,120,123,125,126,127,128,129,132,134,135,139,143,151,152,153,154,156,157,158,160,163,165,166,168,169,173,182,187,190,191,192,193,194]
ZigBee	[5,32,39,42,63,65,66,67,93,119,120,121,122,123,128,133,137,144,151,157,175,185,191]
Bluetooth	[46,53,81,118,129,132,134,156,174,181,195,196]
LoRa	[14,31,78,95,142,144,145,149,157,196,197]
GPRS	[44,45,67,73,79,98,101,122,137,157,161,171,179,186,190]
MQTT	[47,75,86,113,114,160,175,176]
4G	[14,45,78,138,141,145,152]
6LoWPAN	[75,147]
IEEE 802.15.4	[73,102]

**Table 7 sensors-20-01042-t007:** Most utilized communication modules.

Technology	References
ESP82666	[34,43,47,56,64,69,71,82,84,86,87,94,97,111,113,114,116,119,123,126,127,129,158,160,166,168,169,177,182,187,193]
SIM900	[35,36,44,61,72,122,129,179,192]
NRF24L01	[124,131,173]
XBee S2	[119,121,144]
SX1276	[78,95,145]

**Table 8 sensors-20-01042-t008:** Cloud platforms.

Cloud	References
Unspecified Cloud	[14,32,41,50,64,72,77,78,88,99,110,113,114,119,121,132,142,151,152,153,156,169,175,197]
Thingspeak	[44,56,98,115,118,130,143,146,158,160,166,170,173,191]

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
