# Peer review of "IoT-Based Smart Irrigation Systems: An Overview on the Recent Trends on Sensors and IoT Systems for Irrigation in Precision Agriculture"

_sensors, 2020, doi:10.3390/s20041042_

Round 1

Reviewer 1 Report

The authors have identified a good and noteworthy topic of irrigation system and presented a comprehensive summary of diverse irrigation systems implemented across the world.

This manuscript does a good job of identifying the most common architecture, components, software, and technology used in irrigation systems. While the presented summary is comprehensive of information from the cited papers, limiting the scope of this work to just irrigation systems, but not agriculture systems is one of the fundamental limitations. Extensive published research has shown that the agricultural yield can be improved by using methods beyond irrigation systems. Additionally, many systems beyond irrigation are required for farming and agriculture.

The majority of this manuscript presents a summary of work done by others, with very limited or no new intellectual merits. For instance, an extensive discussion of the pros and cons of common architectures, components, software, and technology used in irrigation systems is lacking.

Security and privacy are a rising concern in smart systems. However, this manuscript does not present any information on security and privacy issues, recent work done in this area, and how irrigation systems could be improved to resolve these issues.

Similarly, what are the issues with big data that is being collected, and related analytics?

One of the major limitations in IoT systems is power consumption in data transmission. This paper states that the TCP/IP protocol is to be used for irrigation systems. However, significant research shows that MQTT results in lower power consumption for IoT devices. This manuscript does not account for any of this information.

Additionally, what are the future directions or challenges in irrigation systems?

Overall, this manuscript does present a decent summary of information from several papers but lacks scientific merit in terms of discussion and analysis.

Author Response

RESPONSE FOR REVIEWERS:

We really thank the reviewers for their very useful comments. The following modifications have been made in the revision.

Reviewer 1

The authors have identified a good and noteworthy topic of irrigation system and presented a comprehensive summary of diverse irrigation systems implemented across the world.

Comment 1: This manuscript does a good job of identifying the most common architecture, components, software, and technology used in irrigation systems. While the presented summary is comprehensive of information from the cited papers, limiting the scope of this work to just irrigation systems, but not agriculture systems is one of the fundamental limitations. Extensive published research has shown that the agricultural yield can be improved by using methods beyond irrigation systems. Additionally, many systems beyond irrigation are required for farming and agriculture.

Reply 1: Certainly, there are more methods apart from irrigation that help improving agricultural yield. However, if this survey regarded all papers available for agriculture systems, the scope of the survey would be too wide and difficult to cover. Furthermore, the result would surpass a length of 100 pages and could be enough for publishing a book. There are, however, other surveys on precision agriculture and specific aspects of irrigation techniques that are available. As explained on the fifth paragraph of the introduction, with this paper we aim to address the current gap in literature of surveys on the current state of the art of IoT-based smart irrigation systems.

Comment 2: The majority of this manuscript presents a summary of work done by others, with very limited or no new intellectual merits. For instance, an extensive discussion of the pros and cons of common architectures, components, software, and technology used in irrigation systems is lacking.

Reply 2: We have added new comments to the different sections of the survey as well as to the discussion section on these topics.

Regarding the components, Sections 3, 4 and 5 depict the most utilized sensors and actuators for the monitoring activities performed by irrigation systems. Sections 3 and 4 present a more extended commentary on the characteristics on the sensors. Therefore, we have extended the discussion on the sensors in Section 5 with the following text (See pages 14 and 16):

“On the one hand, the DHT11 and DHT22 provide both temperature and relative humidity readings. On the other hand, the LM35 and the TMP-36 have broader temperature ranges.”

“The DHT11 and DHT22 are low-cost sensors. However, the AM2315 has a higher price with the advantages of a higher range (-40ºC to 125ºC) has higher accuracy (±0.1ºC).”

“The SH10 and SH11 sensors provide both temperature and relative humidity readings. Furthermore, compared to the DHT11, they present better temperature ranges and accuracy. They are however similar to the DHT22 in terms of temperature ranges, accuracy and price.”

“The DHT22, AM2315, and SHT10 have the same range. However, the AM2315 has higher accuracy (±2%) than that of the DHT22 and the SH10 (±5% and ± 4.5% respectively).”

“Among them, the BHT1750 is the sensor with a broader range (1-65535 Lux).”

Section 3 has been extended as well to add information on evapotranspiration (See page 10).

“In addition to the defined parameters, which are measured in different papers included in this survey, there is an important parameter that is crucial for irrigation scheduling. This parameter indicates the water loss due to the evaporation from the soil and transpiration through the stomata of plants; it is known as evapotranspiration (ET). Several authors measured both terms separately, evaporation and transpiration, using different sensors to have the data of ET [105]. It also can be estimated according to the vegetation and the climatic data by several mathematic models [106]. Some web-based applications can estimate the potential ET given a location. For ET monitoring, remote sensing is the best option as many papers point [107-109]. Among the possibilities for ET monitoring with sensors, the integration of soil moisture changes can offer data of evaporation. Nonetheless, the measurement of transpiration is more complex, it can be estimated with the data of sensors that measure radiation, leaf area index, and vapor pressure deficit [110]. Due to the complexity of its measurement, it is not included as a monitored variable in most of the papers of precision agriculture. No one of the papers considered by this survey includes all the necessary sensors for measuring the ET.”

The most utilized nodes are depicted in section 6.1. The discussion on this section has been extended as well (See page 20):

“The selection of the best node for an IoT irrigation system will depend on the necessities and the characteristics the farmer wants for the system. Arduino nodes and similar nodes from other brands provide a low-cost solution that can be implemented in developing countries and smaller farms. On the other hand, Raspberries have powerful computing abilities that allow the implementation of more demanding software and algorithms.”

“The proposals that opt for developing their own designs for the nodes aim at addressing their own particular requirements. Therefore, the selection of the processor would depend on the characteristics of the IoT irrigation system considering the type of crop and its irrigation needs.”

Communication technologies are depicted in section 6.2. The discussion of the presented technologies has been extended (See pages 21 and 22):

“Therefore, range, data rate, and energy consumption are some of the most important aspects to consider when deciding which technology to use.”

“The reason could be due to is accessibility. The currently available low-cost devices for IoT usually support WiFi and, although its range can be considered short for the average expanse of a farm, small farms could be able to provide enough wireless coverage with several low-cost devices. GSM and ZigBee are widely spread wireless technologies as well, with 25 and 23 papers that use them respectively. GSM provides long-range communication at the cost of a mobile plan of the service provider that operates in the area. ZigBee provides low energy consumption and allows implementing networks with up to 65.000 nodes. However, it has lower data rates than other available technologies and its range would imply the deployment of many nodes. Lastly, there are two new technologies that have been getting popular recently. LoRa is able to provide very long ranges, which makes this technology a very good option for secluded areas with no service. Moreover, regarding specific protocols for IoT systems, it is a little bit surprising that even though MQTT is a widely spread protocol due to its low power consumption and low overhead, it is not a popular protocol for IoT irrigation systems at the moment.”

The cloud platforms used to manage these systems are presented in section 6.3. The pros and cons of the software presented in this section have been added as well (See page 29):

“This platform is very intuitive and provides both free and paid options for storing, analyzing and displaying the data on different devices. Algorithms can be developed using MATLAB to generate alerts.”

“These less-used platforms are either more expensive, provide fewer services or are less intuitive than Thingspeak”

Lastly, the discussion on the architectures provided in section 7.7 has been improved (See page 34):

“Traditionally, the IoT architecture has been considered to be divided into three layers, which are the perception, network, and application layer. Later on, an intermediate layer placed between the network and application layers has been introduced in multiple studies. This layer, called Service Layer, is employed to store and process data using cloud and fog computing. For the last few years, authors such as Ferrández-Pastor [217] presented a new architecture proposal, based on four layers: Things, Edge, Communication, and Cloud. In these latest architectural proposals, the authors use the Edge layer to locate critical applications and perform basic control processes. Also, as indicated by [217], Cloud (Internet/intranet) can provide Web services, data storage, HMI interfaces or analytic applications. Figure 29 shows an image where you can see the architecture models.”

Comment 3: Security and privacy are a rising concern in smart systems. However, this manuscript does not present any information on security and privacy issues, recent work done in this area, and how irrigation systems could be improved to resolve these issues.

Reply 3: The information on security in IoT systems for irrigation is presented in section 7.6. We have, extended this section according to the provided suggestions (See pages 32-34):

“On the one hand, all the security issues of any IoT system can be applied to an IoT irrigation system for agriculture. Therefore, current works with their focus on securing IoT describes the security challenges that IoT irrigation systems may face. For example, organizations such as the Internet Engineering Task Force (IETF) study the different problems that affect.”

“However, on the other hand, when considering just the currently available studies on securing IoT irrigation systems specifically, it is possible to discern which aspects of IoT security are prioritized for this type of system.”

“Another threat to the devices deployed on the field is the possibility of them being replaced to malicious nodes, providing the attacker with access to the network [210]. The deployed devices may also be susceptible to malicious code and false data injection, leading to wrong results and the malfunction of the system. Sleep deprivation attacks are aimed at depleting the battery of the devices. The deployed nodes are susceptible to booting attacks as well. Furthermore, attackers may also perform eavesdropping and interfere with the deployed devices. The concern of securing the physical devices deployed on the fields has led to many systems incorporating PIR sensors, scarecrows, and buzzers to detect intruders, either humans or animals, and to alert of their presence [58, 211].

Privacy is another aspect to consider. The data managed by IoT irrigation systems may not need as much privacy as the data managed by other IoT applications such as those for health.”

“which is avoiding DoS attacks. However, the owners of the fields may be implementing new techniques they would prefer to remain private or the water quality management system could be regarded as a critical infrastructure as the produce that results from the fields irrigated with that water would be consumed by people. Therefore, considering this aspect, Ahad et al. [204] and Barrento et al. [212] remark the need for user privacy, as the obtention of this information may lead to attacks to the owner or the personnel of the farm. Ahad et al. [204] consider data and device privacy as well, because of the need for guaranteeing the ownership of the data to avoid repudiation and ensuring data availability. Barreto et al. [212] remark the burden that could be originated from the exposure of the GPS locational data captured by IoT devices as attackers may gain information on the location of the farm to perform physical attacks. Lastly, Ahad et al. [204] also comment on the need for a secure storage system for the information, with particular emphasis on distributed data storage systems. Attacks such as SQL injections can lead to the obtention of private information from storage systems [210].

As cloud computing often goes hand in hand with IoT systems, threats to cloud services may compromise the IoT irrigation system as well. Flooding attacks and cloud malware injection are some of the attacks that can be intentionally executed to compromise the data and the performance of the cloud [210].

Ransomware is another security threat that could affect irrigation systems for agriculture [212]. All the data regarding pesticides and the fertigation system could be encrypted until a ransom is paid. For this reason, having a back-up in a remote location is greatly advised.

Barreto et al. [212] also remark some security threats that are not usually commented on when discussing the security of IoT systems for agriculture. One of these threats is the damage that can be caused by social engineering as the users of the IoT irrigation system could provide login information to people posing as technicians or click on malicious links on their e-mails. The other threats are agroterrorism, cyber-espionage, mostly related to accessing intellectual property or confidential information regarding aspects such as genetically modified crops.

Blockchain is a new technique that is being applied to secure IoT systems [210], which allows secure data storage and communication. In agriculture, it is mostly utilized to secure the supply chain [213], although there are other proposals that use it to secure a greenhouse [214] or to secure the overall smart agriculture framework [215]. Regarding IoT irrigation systems for agriculture, blockchain has been applied in [216] to track and trace the information exchange of their proposed smart watering system.”

There aren’t many papers solely focused on the security of IoT systems for irrigation. Most papers focus on securing IoT systems in general. Therefore, we have expanded on the specific aspects that can affect IoT irrigation systems.

Comment 4: Similarly, what are the issues with big data that is being collected, and related analytics?

Reply 4: The scope of this survey is primarily focused on the sensors used in irrigation systems. However, as these topics are greatly important, we have added some information on these topics on the discussion section modifying Section 7.1. (See page 30):

“IoT systems, in general, generate a great amount of data due to monitoring different parameters in real-time, and IoT irrigation systems generate big data as well. Therefore, it is necessary to provide mechanisms to manage and analyze big data. Ahad et al. [204] comment on the need for sustainable management of big data to avoid overusing natural resources. Using blockchain, powering IoT devices with solar energy, selecting useful data and discarding unnecessary and redundant data, employing clustering techniques to reduce the volume of the information, implementing time-efficient algorithms and the utilization of sustainable resources as alternatives to the components that are employed nowadays are some of the suggestions they provide.

Although the data gathered from the sensors is already a big source of information, the analysis of this data is paramount to optimize the IoT irrigation system according to the crop and the weather conditions. For this purpose, Tseng et al. [205] presented a big data analysis algorithm to aid farmers in the crop selection process. Their proposal satisfied the 5V (volume, velocity, variety, veracity, and value) of big data. They performed a 3D correlation analysis that evaluated the irrigation cycle to identify the irrigation practices performed by the farmer. Then, they calculated the soil moisture content to detect irrigation and to evaluate if the farmer applied pesticides or fertilizers. This information was coupled with data on the weather conditions and other aspects of the soil to provide the farmer with the cultivation risks of each type of crop. Nonetheless, there are other technologies that are currently being utilized to analyze the big data produced by IoT irrigation systems for agriculture.”

Comment 5: One of the major limitations in IoT systems is power consumption in data transmission. This paper states that the TCP/IP protocol is to be used for irrigation systems. However, significant research shows that MQTT results in lower power consumption for IoT devices. This manuscript does not account for any of this information.

Reply 5: We have commented on the use of MQTT on irrigation systems in Section 6.2 (See pages 22 and 26):

“Moreover, regarding specific protocols for IoT systems, it is a little bit surprising that even though MQTT is a widely spread protocol due to its low power consumption and low overhead, it is not a popular protocol for IoT irrigation systems at the moment.”

“MQTT  [45, 75, 86, 113, 114, 162, 177, 178]”

Comment 6: Additionally, what are the future directions or challenges in irrigation systems?

Reply 6: The future directions and challenges in irrigation systems have been added in section 7.9 (See pages 36 and 37):

“Internet of Underground Things (IoUT) is a new view on IoT [223]. It consists of deploying both underground and above ground IoT devices that communicate among themselves with underground-underground, underground-above ground, and above ground-above ground communication. It is especially useful and applicable to irrigation and precision agriculture IoT systems as the devices would not be impeding the work of machines and farmers and it would reduce the amount of physical damage the nodes deployed on the fields may receive. Although WiFi surprisingly allows underground-above ground communication in short distances and depths above 30 cm [224], it would be necessary to study the performance of other existing protocols or even the creation of a new protocol that employs lower frequency bands to transmit the information through the soil medium.

The use of LoRa is increasing for irrigation and precision agriculture applications due to its long-range, which allows wireless communication to remote fields. However, the advances in 5G may lead to a decrease in the interest in LoRa technology. On the one hand, if the designed irrigation system produces lower amounts of data, as these types of systems present low variability in the data, LoRa presents itself as a very good solution. However, on the other hand, for IoT systems that require the transmission of large amounts of data, 5G would solve the problem of the limited amount of data that LoRa can transmit. Considering these aspects, a possible future is the use of LoRa and 5G Hybrid Wireless Networks to attempt in satisfying those needs [225].

There are diverse opinions on the existence and effects of climate change on earth. However, there is no doubt that many governments, companies and the citizens themselves are increasing their awareness regarding sustainable consumption. This not only affects their view on the energy consumption of IoT systems for irrigation but also, the origin of the components of the devices and the impact of IoT devices on the environment and the fauna of the area where they are deployed. Therefore, the reduction in energy consumption and the use of alternative powering methods will continue to be a research trend. Furthermore, it may be a challenge to find new materials that are weather-resistant and do not increase severely the cost of the devices, but the use of recyclable materials in some of the elements of the IoT devices is to be expected [204].

As it has further been commented on subsection 7.1, blockchain and AI will be incorporated into most IoT services [210], including those aimed at irrigation and precision agriculture. These technologies provide not only increased security but also optimization to the management of the IoT systems. This will lead to an increased understanding of the crops and a faster way to characterize genetically modified crops and the effects of new fertilizers and pesticides. Furthermore, it will aid in optimizing and reducing water consumption and improving the new mechanisms that are being proposed to determine if treated wastewater can be used for irrigation and which crops can be irrigated with wastewater depending on its composition.

Lastly, farmers usually have a very small profit and thus, these IoT systems may not be affordable to them. Therefore, the cost of IoT devices and the overall system implementation should have a decreasing tendency for these types of systems to have a commercial future.”

Reviewer 2 Report

The article presents a review on the Recent Trends of the Sensors and IoT Systems for Irrigation in Precision Agriculture.

The objective is to analyze the current parameters that are monitored in irrigation systems regarding water quantity and quality, soil characteristics and weather conditions, and an overview of the most utilized nodes and wireless technologies.

The work carries out an extensive review of the technologies applied in the scope of the review.  However, it does not include the latest advances in intelligent systems and Precision Agriculture architectures.

Figures 29 and 30 are standard architectures that are being expanded by more complex proposals where IoT systems and AI paradigms are integrated. Examples such as  https://doi.org/10.3390/s16071141 or similar show these latest proposals. Current intelligent systems also integrate new programming paradigms such as edge or fog computing. Cloud computing, machine learning ,deep-learning ... also begin to be used in these scenarios. Prediction systems for irrigation, pattern recognition are examples of new trends to optimize water use

To improve the review of this work, the latest advances and proposals mentioned above must be considered.

Author Response

We really thank the reviewers for their very useful comments. The following modifications have been made in the revision.

Reviewer 2

The article presents a review on the Recent Trends of the Sensors and IoT Systems for Irrigation in Precision Agriculture.

The objective is to analyze the current parameters that are monitored in irrigation systems regarding water quantity and quality, soil characteristics and weather conditions, and an overview of the most utilized nodes and wireless technologies.

Comment 1: The work carries out an extensive review of the technologies applied in the scope of the review.  However, it does not include the latest advances in intelligent systems and Precision Agriculture architectures.

Reply 1: As it is stated in Section 2, we only considered papers that were related or discussed irrigation. However, we have expanded on this this topic in Section 7 and more intensively in subsection 7.9.

On Big data management and analytics (See page 30, subsection 7.1):

“IoT systems, in general, generate a great amount of data due to monitoring different parameters in real-time, and IoT irrigation systems generate big data as well. Therefore, it is necessary to provide mechanisms to manage and analyze big data. Ahad et al. [204] comment on the need for sustainable management of big data to avoid overusing natural resources. Using blockchain, powering IoT devices with solar energy, selecting useful data and discarding unnecessary and redundant data, employing clustering techniques to reduce the volume of the information, implementing time-efficient algorithms and the utilization of sustainable resources as alternatives to the components that are employed nowadays are some of the suggestions they provide.

Although the data gathered from the sensors is already a big source of information, the analysis of this data is paramount to optimize the IoT irrigation system according to the crop and the weather conditions. For this purpose, Tseng et al. [205] presented a big data analysis algorithm to aid farmers in the crop selection process. Their proposal satisfied the 5V (volume, velocity, variety, veracity, and value) of big data. They performed a 3D correlation analysis that evaluated the irrigation cycle to identify the irrigation practices performed by the farmer. Then, they calculated the soil moisture content to detect irrigation and to evaluate if the farmer applied pesticides or fertilizers. This information was coupled with data on the weather conditions and other aspects of the soil to provide the farmer with the cultivation risks of each type of crop. Nonetheless, there are other technologies that are currently being utilized to analyze the big data produced by IoT irrigation systems for agriculture.”

On security (See pages 32-34, subsection 7.6):

“On the one hand, all the security issues of any IoT system can be applied to an IoT irrigation system for agriculture. Therefore, current works with their focus on securing IoT describes the security challenges that IoT irrigation systems may face. For example, organizations such as the Internet Engineering Task Force (IETF) study the different problems that affect.”

“However, on the other hand, when considering just the currently available studies on securing IoT irrigation systems specifically, it is possible to discern which aspects of IoT security are prioritized for this type of system.”

“Another threat to the devices deployed on the field is the possibility of them being replaced to malicious nodes, providing the attacker with access to the network [210]. The deployed devices may also be susceptible to malicious code and false data injection, leading to wrong results and the malfunction of the system. Sleep deprivation attacks are aimed at depleting the battery of the devices. The deployed nodes are susceptible to booting attacks as well. Furthermore, attackers may also perform eavesdropping and interfere with the deployed devices. The concern of securing the physical devices deployed on the fields has led to many systems incorporating PIR sensors, scarecrows, and buzzers to detect intruders, either humans or animals, and to alert of their presence [58, 211].

Privacy is another aspect to consider. The data managed by IoT irrigation systems may not need as much privacy as the data managed by other IoT applications such as those for health.”

“which is avoiding DoS attacks. However, the owners of the fields may be implementing new techniques they would prefer to remain private or the water quality management system could be regarded as a critical infrastructure as the produce that results from the fields irrigated with that water would be consumed by people. Therefore, considering this aspect, Ahad et al. [204] and Barrento et al. [212] remark the need for user privacy, as the obtention of this information may lead to attacks to the owner or the personnel of the farm. Ahad et al. [204] consider data and device privacy as well, because of the need for guaranteeing the ownership of the data to avoid repudiation and ensuring data availability. Barreto et al. [212] remark the burden that could be originated from the exposure of the GPS locational data captured by IoT devices as attackers may gain information on the location of the farm to perform physical attacks. Lastly, Ahad et al. [204] also comment on the need for a secure storage system for the information, with particular emphasis on distributed data storage systems. Attacks such as SQL injections can lead to the obtention of private information from storage systems [210].

As cloud computing often goes hand in hand with IoT systems, threats to cloud services may compromise the IoT irrigation system as well. Flooding attacks and cloud malware injection are some of the attacks that can be intentionally executed to compromise the data and the performance of the cloud [210].

Ransomware is another security threat that could affect irrigation systems for agriculture [212]. All the data regarding pesticides and the fertigation system could be encrypted until a ransom is paid. For this reason, having a back-up in a remote location is greatly advised.

Barreto et al. [212] also remark some security threats that are not usually commented on when discussing the security of IoT systems for agriculture. One of these threats is the damage that can be caused by social engineering as the users of the IoT irrigation system could provide login information to people posing as technicians or click on malicious links on their e-mails. The other threats are agroterrorism, cyber-espionage, mostly related to accessing intellectual property or confidential information regarding aspects such as genetically modified crops.

Blockchain is a new technique that is being applied to secure IoT systems [210], which allows secure data storage and communication. In agriculture, it is mostly utilized to secure the supply chain [213], although there are other proposals that use it to secure a greenhouse [214] or to secure the overall smart agriculture framework [215]. Regarding IoT irrigation systems for agriculture, blockchain has been applied in [216] to track and trace the information exchange of their proposed smart watering system.”

On future directions and challenges in irrigation systems (See pages 36 and 37, subsection 7.9):

“Internet of Underground Things (IoUT) is a new view on IoT [223]. It consists of deploying both underground and above ground IoT devices that communicate among themselves with underground-underground, underground-above ground, and above ground-above ground communication. It is especially useful and applicable to irrigation and precision agriculture IoT systems as the devices would not be impeding the work of machines and farmers and it would reduce the amount of physical damage the nodes deployed on the fields may receive. Although WiFi surprisingly allows underground-above ground communication in short distances and depths above 30 cm [224], it would be necessary to study the performance of other existing protocols or even the creation of a new protocol that employs lower frequency bands to transmit the information through the soil medium.

The use of LoRa is increasing for irrigation and precision agriculture applications due to its long-range, which allows wireless communication to remote fields. However, the advances in 5G may lead to a decrease in the interest in LoRa technology. On the one hand, if the designed irrigation system produces lower amounts of data, as these types of systems present low variability in the data, LoRa presents itself as a very good solution. However, on the other hand, for IoT systems that require the transmission of large amounts of data, 5G would solve the problem of the limited amount of data that LoRa can transmit. Considering these aspects, a possible future is the use of LoRa and 5G Hybrid Wireless Networks to attempt in satisfying those needs [225].

There are diverse opinions on the existence and effects of climate change on earth. However, there is no doubt that many governments, companies and the citizens themselves are increasing their awareness regarding sustainable consumption. This not only affects their view on the energy consumption of IoT systems for irrigation but also, the origin of the components of the devices and the impact of IoT devices on the environment and the fauna of the area where they are deployed. Therefore, the reduction in energy consumption and the use of alternative powering methods will continue to be a research trend. Furthermore, it may be a challenge to find new materials that are weather-resistant and do not increase severely the cost of the devices, but the use of recyclable materials in some of the elements of the IoT devices is to be expected [204].

As it has further been commented on subsection 7.1, blockchain and AI will be incorporated into most IoT services [210], including those aimed at irrigation and precision agriculture. These technologies provide not only increased security but also optimization to the management of the IoT systems. This will lead to an increased understanding of the crops and a faster way to characterize genetically modified crops and the effects of new fertilizers and pesticides. Furthermore, it will aid in optimizing and reducing water consumption and improving the new mechanisms that are being proposed to determine if treated wastewater can be used for irrigation and which crops can be irrigated with wastewater depending on its composition.

Lastly, farmers usually have a very small profit and thus, these IoT systems may not be affordable to them. Therefore, the cost of IoT devices and the overall system implementation should have a decreasing tendency for these types of systems to have a commercial future.”

Comment 2: Figures 29 and 30 are standard architectures that are being expanded by more complex proposals where IoT systems and AI paradigms are integrated. Examples such as  https://doi.org/10.3390/s16071141 or similar show these latest proposals. Current intelligent systems also integrate new programming paradigms such as edge or fog computing. Cloud computing, machine learning ,deep-learning ... also begin to be used in these scenarios. Prediction systems for irrigation, pattern recognition are examples of new trends to optimize water use

Reply 2: We have expanded on the mentioned topics on the discussion section and we have updated Figure 29 (See page 34, section 7.7):

“Traditionally, the IoT architecture has been considered to be divided into three layers, which are the perception, network, and application layer. Later on, an intermediate layer placed between the network and application layers has been introduced in multiple studies. This layer, called Service Layer, is employed to store and process data using cloud and fog computing. For the last few years, authors such as Ferrández-Pastor [217] presented a new architecture proposal, based on four layers: Things, Edge, Communication, and Cloud. In these latest architectural proposals, the authors use the Edge layer to locate critical applications and perform basic control processes. Also, as indicated by [217], Cloud (Internet/intranet) can provide Web services, data storage, HMI interfaces or analytic applications. Figure 29 shows an image where you can see the architecture models.”

Furthermore, on the topics of machine-learning, predictions and pattern recognition, we have extended section 7.1 (See page 30):

“IoT systems, in general, generate a great amount of data due to monitoring different parameters in real-time, and IoT irrigation systems generate big data as well. Therefore, it is necessary to provide mechanisms to manage and analyze big data. Ahad et al. [204] comment on the need for sustainable management of big data to avoid overusing natural resources. Using blockchain, powering IoT devices with solar energy, selecting useful data and discarding unnecessary and redundant data, employing clustering techniques to reduce the volume of the information, implementing time-efficient algorithms and the utilization of sustainable resources as alternatives to the components that are employed nowadays are some of the suggestions they provide.

Although the data gathered from the sensors is already a big source of information, the analysis of this data is paramount to optimize the IoT irrigation system according to the crop and the weather conditions. For this purpose, Tseng et al. [205] presented a big data analysis algorithm to aid farmers in the crop selection process. Their proposal satisfied the 5V (volume, velocity, variety, veracity, and value) of big data. They performed a 3D correlation analysis that evaluated the irrigation cycle to identify the irrigation practices performed by the farmer. Then, they calculated the soil moisture content to detect irrigation and to evaluate if the farmer applied pesticides or fertilizers. This information was coupled with data on the weather conditions and other aspects of the soil to provide the farmer with the cultivation risks of each type of crop. Nonetheless, there are other technologies that are currently being utilized to analyze the big data produced by IoT irrigation systems for agriculture.”

Reviewer 3 Report

The authors investigate an interesting and important problem: providing sensing and decision support for data-driven water management for agriculture. The authors perform a research paper survey of 178 papers and categorize the contributions of each in terms of location, water quantity and quality, soil characteristics, weather conditions, sensor types, use, and characteristics, communications technologies, and other features and methods employed. The authors report on the number of papers that focus or advance each of and discuss the similarities and differences between them.

Although this survey is very comprehensive and useful, I do not feel that it is a good fit for this Journal. This is because there are very few new techniques, insights or advances proposed (besides a couple of paragraphs on recommendations). Perhaps a venue that focuses on surveys would be a better fit.

The paper is pretty well written and the topic/review coverage is extensive. The paper is useful more as a reference rather than a research advance in sensing, decision support, and/or analytics for precision irrigation. The one topic that appears to be missing however is evapotranspiration (various ways of computing/estimating/validating it) and its use to guide precision and deficit irrigation.  It is only mentioned in a couple of places but is used extensively in this area.

Some improvements can also be made to the writing. For example, there are paragraphs that "run-on" for a page or more. These can be divided into topics to improve the paper.

Author Response

We really thank the reviewers for their very useful comments. The following modifications have been made in the revision.

Reviewer 3

The authors investigate an interesting and important problem: providing sensing and decision support for data-driven water management for agriculture. The authors perform a research paper survey of 178 papers and categorize the contributions of each in terms of location, water quantity and quality, soil characteristics, weather conditions, sensor types, use, and characteristics, communications technologies, and other features and methods employed. The authors report on the number of papers that focus or advance each of and discuss the similarities and differences between them.

Comment 1: Although this survey is very comprehensive and useful, I do not feel that it is a good fit for this Journal. This is because there are very few new techniques, insights or advances proposed (besides a couple of paragraphs on recommendations). Perhaps a venue that focuses on surveys would be a better fit.

Reply 1: This paper has been forwarded to a special issue on surveys. However, we have expanded the discussion section to comment more in detail on the advances in IoT irrigation systems (See Section 7).

Comment 2: The paper is pretty well written and the topic/review coverage is extensive. The paper is useful more as a reference rather than a research advance in sensing, decision support, and/or analytics for precision irrigation. The one topic that appears to be missing however is evapotranspiration (various ways of computing/estimating/validating it) and its use to guide precision and deficit irrigation.  It is only mentioned in a couple of places but is used extensively in this area.

Reply 2: We have added new content on evapotranspiration (See Section 3, page 10):

“In addition to the defined parameters, which are measured in different papers included in this survey, there is an important parameter that is crucial for irrigation scheduling. This parameter indicates the water loss due to the evaporation from the soil and transpiration through the stomata of plants; it is known as evapotranspiration (ET). Several authors measured both terms separately, evaporation and transpiration, using different sensors to have the data of ET [105]. It also can be estimated according to the vegetation and the climatic data by several mathematic models [106]. Some web-based applications can estimate the potential ET given a location. For ET monitoring, remote sensing is the best option as many papers point [107-109]. Among the possibilities for ET monitoring with sensors, the integration of soil moisture changes can offer data of evaporation. Nonetheless, the measurement of transpiration is more complex, it can be estimated with the data of sensors that measure radiation, leaf area index, and vapor pressure deficit [110]. Due to the complexity of its measurement, it is not included as a monitored variable in most of the papers of precision agriculture. No one of the papers considered by this survey includes all the necessary sensors for measuring the ET.”

Comment 3: Some improvements can also be made to the writing. For example, there are paragraphs that "run-on" for a page or more. These can be divided into topics to improve the paper.

Reply 3: The text has been checked to avoid run-on paragraphs.

Round 2

Reviewer 1 Report

Good job on addressing the concerns raised in the initial review. The quality of the article has improved significantly.

The manuscript could use one additional proof reading for typographical errors (eg. page numbers) and grammatical mistakes.

Overall, a good job.

Reviewer 2 Report

The work has been expanded and improved. Congratulations for the work done

Reviewer 3 Report

The authors made significant improvements to this version. The writing has been improved.  I thank the authors for their efforts.

This paper is a high-quality survey of a particular subfield of precision ag.  I still do not feel that this is appropriate for Sensors (given there are no new insights or novel extensions/uses over any of this existing work) but as a survey this is a strong paper.